# Wdr47, Camsaps, and Katanin cooperate to generate ciliary central microtubules

Hao Liu[1,2,7], Jianqun Zheng[1,2,7], Lei Zhu[1,2,7], Lele Xie[1], Yawen Chen[1,2], Yirong Zhang[1,2], Wei Zhang[1], Yue Yin[3], Chao Peng [3], Jun Zhou [4], Xueliang Zhu [1,2,5,8✉] & Xiumin Yan[1,6,8✉]

The axonemal central pair (CP) are non-centrosomal microtubules critical for planar ciliary beat. How they form, however, is poorly understood. Here, we show that mammalian CP formation requires Wdr47, Camsaps, and microtubule-severing activity of Katanin. Katanin severs peripheral microtubules to produce central microtubule seeds in nascent cilia. Camsaps stabilize minus ends of the seeds to facilitate microtubule outgrowth, whereas Wdr47 concentrates Camsaps into the axonemal central lumen to properly position central microtubules. *Wdr47* deficiency in mouse multicilia results in complete loss of CP, rotatory beat, and primary ciliary dyskinesia. Overexpression of Camsaps or their microtubule-binding regions induces central microtubules in *Wdr47*$^{-/-}$ ependymal cells but at the expense of low efficiency, abnormal numbers, and wrong location. Katanin levels and activity also impact the central microtubule number. We propose that Wdr47, Camsaps, and Katanin function together for the generation of non-centrosomal microtubule arrays in polarized subcellular compartments.

[1] State Key Laboratory of Cell Biology, Shanghai Institute of Biochemistry and Cell Biology, Center for Excellence in Molecular Cell Science, Chinese Academy of Sciences, 320 Yueyang Road, 200031 Shanghai, China. [2] University of Chinese Academy of Sciences, 100049 Beijing, China. [3] National Facility for Protein Science in Shanghai, Zhangjiang Lab, Shanghai Advanced Research Institute, Chinese Academy of Sciences, 201210 Shanghai, China. [4] Institute of Biomedical Sciences, College of Life Sciences, Key Laboratory of Animal Resistance Biology of Shandong Province, Collaborative Innovation Center of Cell Biology in Universities of Shandong, Shandong Normal University, 88 East Wenhua Road, 250014 Jinan, Shandong, China. [5] School of Life Science, Hangzhou Institute for Advanced Study, University of Chinese Academy of Sciences, 310024 Hangzhou, China. [6] Present address: Ministry of Education-Shanghai Key Laboratory of Children's Environmental Health, Institute of Early Life Health, Xinhua Hospital, Shanghai Jiao Tong University School of Medicine, 200092 Shanghai, China. [7] These authors contributed equally: Hao Liu, Jianqun Zheng, Lei Zhu. [8] These authors jointly supervised this work: Xueliang Zhu, Xiumin Yan. ✉email: xlzhu@sibcb.ac.cn; yanx@sibcb.ac.cn

Microtubules (MTs) are organized into different arrays as cytoskeletons of subcellular structures such as the spindle, neurites, and cilia and as tracks for MT-based molecular motors dynein and kinesin[1–4]. Although the centrosome functions as a major MT-organizing center (MTOC) in most animal cells, non-centrosomal MTs also widely exist and are organized into unique arrays in certain cell types or subcellular compartments[5–7]. For instance, non-centrosomal MTs are constructed into parallel arrays perpendicular to the tissue plane in epithelial cells by anchoring their minus ends to adapters at the apical membrane domain[8,9]. They also form unidirectional and bidirectional bundles respectively in axons and dendrites following neuronal polarization[1]. Recently, calmodulin-regulated spectrin-associated proteins (Camsaps) have been found to stabilize free MT minus ends through direct binding to facilitate the rapid growth of the MT plus ends[10–14]. They function in the organization of non-centrosomal MT arrays in epithelial and neuronal cells[15–19].

The majority of motile cilia, e.g. those on the surface of mammalian trachea and ependyma or forming the tail of sperms, contain a "9 + 2"-type axoneme with a pair of non-centrosomal central (C1; C2) MTs surrounded by nine MT doublets extended from the basal body[20,21]. Such a central pair (CP) of MTs are positioned over the center of the transition zone, with their plus ends pointing to the ciliary tip[22–24]. They are interconnected by bridges constituted by proteins such as Spag16 and coated with distinct repetitive arrays of proteinous projections. The resultant supramolecular structure, the CP apparatus, contacts directly with radial spokes emanated from peripheral MT doublets to coordinate axonemal dynein activities[23–25]. Tracheal and ependymal cilia beat in a back-and-forth, or planar, manner. Their loss of CP or CP-associated proteins, such as the C1-associated Spag6 or C2-associated Hydin, results in an abnormal beat pattern and in humans contributes to primary ciliary dyskinesia (PCD)[3,21,26–30].

Centrosomal MTs are nucleated from and stabilized at their minus ends mainly by the γ-tubulin ring complex[2,31]. Usually, non-centrosomal MTs use MTs released from the centrosome as "seeds"[5,6]. Cilium, however, is a subcellular compartment gated by the transition zone[32,33]. While tubulin dimers are found to enter cilia through the intraflagellar transport (IFT) machinery, no evidence suggests that cytoplasmic MTs can pass the transition zone[34,35]. How the initial seeds of the central MTs are produced is thus an outstanding question. Furthermore, although we have previously reported that the mammalian CP formation requires Spef1-mediated MT stabilization in cilia[36], how the number and the position of the central MTs are precisely controlled is still unknown.

In this work, we demonstrate that the WD40 repeat-containing Wdr47 (also called Nemitin)[15,37], Camsaps, and the MT-severing enzyme Katanin[12,38–40], co-operate to achieve the initial central MT formation in mammalian multicilia. Katanin produces central MT seeds by severing the peripheral MTs. Wdr47 concentrates Camsaps into the central lumen, where Camsaps stabilize the minus ends of the seeds to allow central MT elongation. As emerging evidence suggests that Wdr47, Camsaps, and Katanin also similarly impact neuronal polarization and axonal growth[15,16,18,41,42], our findings indicate that the same pathway is used to properly generate and organize non-centrosomal MT arrays in polarized subcellular compartments such as the axon and motile cilium.

## Results

### Wdr47 is a CP-associated protein implicated in CP MT formation. Data mining in our cDNA array results[43] and

subsequent immunoblotting indicated that Wdr47 (Fig. 1a) was upregulated during multiciliation of mouse tracheal epithelial cells (mTECs) cultured at an air-liquid interface (ALI) (Fig. 1b). Wdr47 was highly expressed in mouse tissues abundant in the 9 + 2 type of cilia, such as the lung, the testis, the ependyma, and the oviduct (Fig. 1c). Consistently, immunostaining of cultured mouse ependymal cells (mEPCs) revealed a ciliary-shaft localization of Wdr47 in multicilia (Fig. 1d). Interestingly, Wdr47 was highly concentrated in short cilia, especially at the ciliary tip. Its ciliary-tip intensities declined following ciliary elongation, accompanied with the emergence of Wdr47-positive puncta along and especially at the bottom of the ciliary shaft above ciliary transition zone (TZ), when Cep162, a protein at the bottom region of TZ[44], was used as marker (Fig. 1d)[45]. By contrast, Wdr47 was not detected in the shaft of primary cilium (Supplementary Fig. 1).

Super-resolution microscopy revealed that the ciliary Wdr47 was distributed at the CP region when Hydin[23,28,29] was used as CP marker, with its bottommost distribution preceding that of Hydin (Fig. 1e). Interestingly, in short cilia decorated strongly with Wdr47, Hydin was sometimes weak or undetectable (Fig. 1e), suggesting that central MTs are either not yet formed or still in initial production stages. In such cilia, however, Wdr47 was still concentrated in the central lumen of axonemes in addition to its tip accumulation (Fig. 1e). When we pre-extracted the cells with Triton X-100 to remove soluble proteins prior to fixation, Wdr47 became undetectable in short cilia but still remained on CP in long cilia (Fig. 1e). Therefore, Wdr47 is a previously undocumented CP-associated protein preferentially enriched at the minus-end region of CP. More importantly, as CP MTs in Chlamydomonas are initially assembled at the ciliary distal region[46], the strong enrichment of Wdr47 in short cilia lacking Hydin strongly implies a role in early stages of CP formation.

### Wdr47 deficiency in mice abolishes CP formation. We generated Wdr47-deficient mice by crossing Wdr47 knockout-first (Wdr47[Kof]/+) mice with EIIa-Cre mice (Fig. 1f)[15] and analyzed the ciliary motility of cultured mEPCs[36]. Comparing to the back-and-forth beat pattern of the Wdr47[+/+] and Wdr47[+/−] multi-cilia, the Wdr47[−/−] multicilia moved in a rotatory manner but with similar frequency (Fig. 1g and Supplementary Movie 1). When fluorescent beads were added to the culture medium, the beating Wdr47[+/+] or Wdr47[+/−] cilia drove rapid directional flows of the beads (Fig. 1h). The rotatory Wdr47[−/−] cilia, however, failed to do so (Fig. 1h).

To clarify whether the rotatory pattern of the Wdr47[−/−] multicilia was due to loss of the central MTs[26,36], we examined the ciliary ultrastructure by transmission EM. While multicilia of Wdr47[+/+] or Wdr47[+/−] mEPCs contained normal 9 + 2 axonemes, those of Wdr47[−/−] mEPCs completely lacked both CP MTs (Fig. 1i). Sometimes electron-dense materials were observed in the central lumen of Wdr47[−/−] cilia (Fig. 1i), similar to CP-free flagella of some Chlamydomonas mutants[47,48]. When ciliary cross-sections (n = 126) were classified into proximal ones, based on the presence of surrounding microvilli[49,50], and distal ones, the luminal materials were observed in 71% of the distal sections and 23% of the proximal ones, possibly due to remnants of CP components. EM analyses indicated that Wdr47[−/−] respiratory multicilia were also CP-less (Supplementary Fig. 2). Wdr47 is thus essential to the CP MTs formation.

### Wdr47 is critical for both ciliary localization of CP proteins and CP maintenance. We observed that the ciliary localizations of multiple CP proteins, including CP MTs-associated Spef1, C2 MT-associated Hydin, C1 MT-associated Spag6, and C1-C2 bridge-

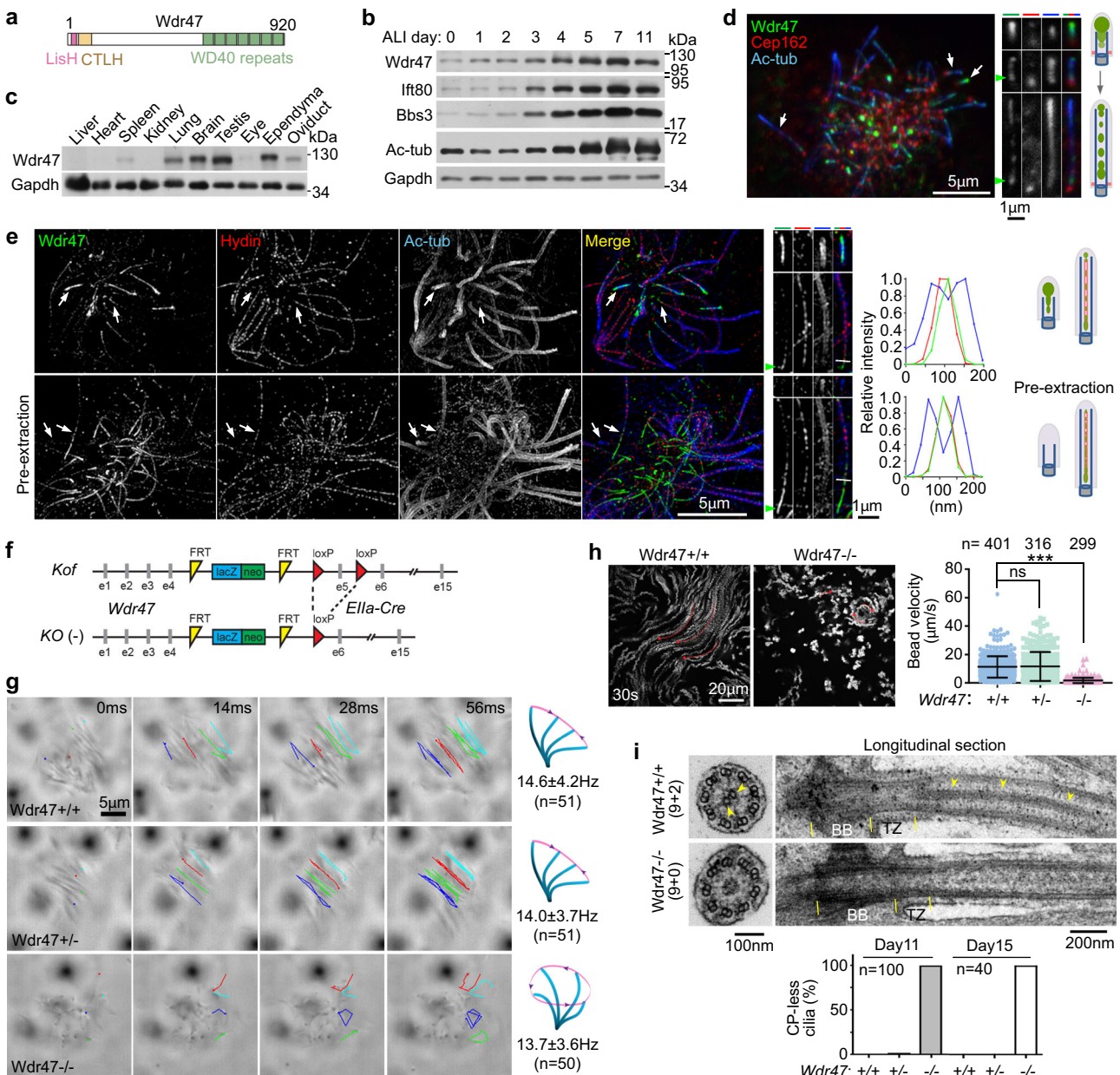

**Fig. 1 Wdr47 is an essential CP protein implicated in initial CP MT formation. a** Diagram for mouse Wdr47. LisH, Lis1 homology; CTLH, carboxyl-terminal to LisH. **b** Upregulation of Wdr47 during multicilia formation. mTECs were cultured at an air–liquid interface (ALI) for the indicated days. Multicilia formation was indicated by the increased levels of Ift80, Bbs3, and acetylated tubulin (Ac-tub) from after ALI d3. Gapdh served as loading control ($n = 2$ biological replicates). **c** Wdr47 was abundant in mouse tissues containing motile cilia. The tissue lysates were prepared from 8-week-old mice ($n = 2$ biological replicates). **d** Wdr47 localized in multicilia as puncta. Cultured mEPCs were serum starved at day 0 to induce multiciliation and fixed at day 7, followed by immunostaining and confocal imaging. Ac-tub and Cep162 labeled the ciliary axoneme and the transition zone (TZ), respectively. Cilia pointed by arrows were magnified by 150% to show distribution changes of Wdr47 following the ciliary elongation. Arrowheads point to Wdr47 puncta at the base of cilia over the TZ. Diagrams are provided to aid comprehension ($n = 5$ biological replicates). **e** Wdr47 entered ciliary central lumen earlier than Hydin in short cilia and was enriched at the bottom region of CP (arrowheads) in long cilia. mEPCs at day 7 were fixed either directly (top panel) or after pre-extraction with Triton X-100 to remove soluble proteins (bottom panel), followed by immunostaining and three dimensional structured illumination microscopy (3D-SIM). Hydin served as CP marker. Cilia pointed by arrows were magnified by 150% to show details. Line scans were performed at positions marked by white lines to show colocalization of Wdr47 with Hydin ($n = 3$ biological replicates). **f** Strategy for generating *Wdr47* knockout (KO) mice. The knockout-first (Kof) allele of *Wdr47* contains a trapping cassette, followed by Exon 5 (e5) flanked by two loxP sites. *Wdr47Kof/+* mice were crossed with *EIIa-Cre* mice to generate *Wdr47+/-* mice, which were then used to produce *Wdr47-/-* mice. *FRT* flippase recognition target, *LacZ* β-galactosidase gene, *neo* neomycin-resistant gene. **g** Representative frames cropped from Supplementary Movie 1 to show trajectories of four cilia in each EPC of the indicated genotypes. The corresponding ciliary beat patterns and ciliary beat frequencies (mean ± s.d.) are also illustrated ($n = 5$ biological replicates). **h** Stacked images showing the multicilia-driven flows of fluorescent beads over 30 s. The arrows indicate flow directions. Beads velocities were quantified from three independent experiments. Data are presented as mean ± s.d. Two-sided Student's *t* test: ns no significance; ***$P < 0.001$. **i** CP MTs (arrowheads) were lost in *Wdr47-/-* ependymal cilia. Cultured mEPCs were fixed at day 11 or day 15 and subjected to transmission EM analyses. BB, basal body (indicated by yellow lines); TZ, transition zone (indicated by yellow lines) ($n = 2$ biological replicates).

associated Spag16 (Fig. 2a and Supplementary Fig. 3)[23,36,51], were abolished or markedly reduced in $Wdr47^{-/-}$ mEPCs, whereas Rsph4a, a radial spoke component[49,52], was not affected (Fig. 2b and Supplementary Fig. 3). To further corroborate this, we purified multicilia from cultured $Wdr47^{+/+}$ and $Wdr47^{-/-}$ mEPCs and performed label-free quantitative (LFQ) proteomic analysis[53]. In addition to Wdr47, 10 CP proteins were hit in the wild-type cilia. Their levels declined by at least 3.7-fold in the $Wdr47^{-/-}$ cilia (Fig. 2c, d). By contrast, subunits of intraflagellar transport (IFT) complexes A and B[4] were either unchanged or only reduced by <2-fold (Fig. 2c, d). Therefore, CP proteins require Wdr47 for their ciliary localization.

To further confirm the importance of Wdr47 in CP formation, we performed rescue experiments by expressing GFP-tagged Wdr47 or Centrin1 (negative control) into $Wdr47^{-/-}$ mEPCs. GFP-Wdr47 entered the $Wdr47^{-/-}$ multicilia and restored the ciliary localization of CP proteins (Fig. 2e and Supplementary Fig. 4). By contrast, the CP proteins still displayed no or weak ciliary localization in the cells expressing Centrin1-GFP (Fig. 2e and Supplementary Fig. 4). Live imaging revealed that multicilia in $Wdr47^{-/-}$ mEPCs expressing Centrin1-GFP were still rotatory, whereas those in $Wdr47^{-/-}$ mEPCs expressing GFP-Wdr47 displayed planar beat (Fig. 2f and Supplementary Movie 2). These results further attribute the lack of CP formation in $Wdr47^{-/-}$ multiciliated cells (Fig. 1i and Supplementary Fig. 2) to the loss of Wdr47.

Next we investigated whether Wdr47 was also required for the stability of pre-existing CP. To do this, we tried to deplete Wdr47 after the CP formation by expressing Cre recombinase in cultured $Wdr47^{flox/flox}$ mEPCs from day −4 or −1 and examined multicilia at day 5 or day 15 (Fig. 2g). The Cre expression at day −4 resulted in rotatory beat pattern and loss of ciliary Hydin even at day 5 (Fig. 2h and Supplementary Fig. 5), suggesting that under the condition Wdr47 was depleted prior to the CP formation in the cells. By contrast, in mEPCs expressing Cre from day −1, Wdr47 was depleted after the CP formation because the incidence of rotatory beat increased dramatically from day 5 to day 15, accompanied with a similar extent of reduction in Hydin-positive multicilia (Fig. 2h and Supplementary Fig. 5). Transmission EM confirmed a 3.5-fold increase of CP-less cilia from day 5 to day 15 (Fig. 2i). Therefore, the maintenance of CP also requires Wdr47.

**Wdr47-deficient mice display PCD-like phenotypes hydrocephalus and sinusitis.** Next, we investigated whether $Wdr47$ deficiency induced multicilia-related pathological disorders in mice. $Wdr47$-deficient mice die immediately after birth due to central nervous system defects[15] and were not suitable for the investigation. The $Wdr47^{Kof/Kof}$ mice, however, are quite heterogeneous in viability: the majority of them die before P55 but the remaining mice live longer than P153[54], probably due to variations in Wdr47 expression levels[15]. To maintain minimal levels of Wdr47 for mouse survival, we created $Wdr47^{Kof/-}$ mice and found that they were viable after birth. $Wdr47^{Kof/-}$ mice were considerably smaller in size than their wild-type littermates (Fig. 3a) and could only survive for <4 weeks. While $Wdr47$-deficient neonatal mice[15] and $Wdr47^{Kof/Kof}$ adult mice[54] only display slightly enlarged brain ventricles, the ventricles of $Wdr47^{Kof/-}$ mice expanded markedly when examined at P14 (Fig. 3b), indicating hydrocephalus[21,28,55]. Moreover, massive amounts of mucus were observed bilaterally in the paranasal cavity of P14 $Wdr47^{Kof/-}$ mice (Fig. 3b), indicating chronic sinusitis[55]. These phenotypes are hallmarks of primary ciliary dyskinesia (PCD)[21,26,56].

To further clarify that the enlarged ventricles were not due to $Wdr47$ deficiency-induced neuronal death[54], we removed the $Kof$

cassette[15] and generated $Wdr47^{flox/flox}$;GFAP-Cre conditional knockout (cKO) mice to specifically knockout $Wdr47$ in GFAP-positive glial cells (Fig. 3c)[57], including EPC progenitors[58,59]. The cKO mice were born normally but developed dome-shaped heads within 3 weeks (Fig. 3d) and died at ~4 weeks old. Coronal brain sections revealed dilated ventricles and magnetic resonance imaging further confirmed severe hydrocephalus of the mice (Fig. 3e). Scanning EM revealed that multicilia in individual ependymal cells tended to cluster together in $Wdr47^{flox/flox}$ mice but became scattered in the cKO mice (Fig. 3f). Live imaging of brain slices revealed planar beat of the $Wdr47^{flox/flox}$ ependymal cilia but rotatory beat of the cKO cilia (Fig. 3g). $Wdr47$ deficiency or insufficiency thus causes PCD.

**Camsaps are CP-associated proteins colocalizing with Wdr47.** How does Wdr47 impact the CP formation? As Wdr47 can be recruited to MTs by MT-binding proteins such as Map8 and Camsap3[15,37], we speculated that it might function with certain MT-binding proteins for the nucleation or stabilization of central MT seeds. Interestingly, we noticed that all three mouse paralogues of Camsaps, Camsap1, Camsap2, and Camsap3, were readily identified from our purified wild-type multicilia samples by mass spectrometry (Fig. 4a). Camsap3 appeared to be the most abundant among the three Camsaps according to both their LFQ intensities and unique peptide counts (Fig. 4a). We have previously shown that all three Camsaps can associate with Wdr47. Moreover, Camsap1 and Camsap3 function as downstream effectors of Wdr47 in neuronal polarization[15]. Therefore, Camsaps might also function in multicilia with Wdr47.

Super-resolution imaging revealed that Camsaps were indeed CP-associated proteins with distribution patterns resembling those of Wdr47. In short cilia negative for Hydin, Camsaps were highly concentrated at the ciliary tip and also distributed in the axonemal central lumen (Fig. 4b). In long cilia they tended to concentrate in the bottom region of the central lumen complementary to Hydin (Fig. 4b). In addition, Camsap1 tended to display punctate distributions along the CP (Fig. 4b). Both endogenous and GFP-tagged Camsaps were distributed above TZ in ependymal cilia when Cep162 and Cep290 were used as TZ markers (Fig. 4c)[44,60]. When mEPCs were pre-extracted with Triton X-100 to remove soluble proteins prior to fixation, the enrichment of Camsaps at the base of the central lumen became more obvious (Fig. 4c). Such a distribution pattern of Camsaps (Fig. 4b, c) resembles proteins localized at the CP minus end region, or the CP-foot[45]. Therefore, similar to Wdr47, Camsaps are preferentially enriched at the tip of nascent short cilia and the minus end region of central MTs in long cilia.

As our antibodies did not allow co-immunostaining with Wdr47, we examined GFP-Camsaps. In mEPCs expressing Centrin1-GFP, immunofluorescent signals of Wdr47 were independent of those of Centrin1-GFP (Fig. 4d), indicating no fluorescent bleed between the two channels. In contrast, GFP-Camsaps co-localized nicely with endogenous Wdr47 in multicilia (Fig. 4d), confirming their interplays with Wdr47 in ependymal cilia.

Although the levels of Camsaps in mEPCs were not affected by the $Wdr47$ deficiency (Fig. 4e), immunostaining confirmed that the ciliary localization of Camsaps was largely dependent on Wdr47 (Fig. 4f, g). Therefore, Camsaps could be downstream effectors of Wdr47 in the CP formation.

**Overexpression of Camsaps induces central MT formation in $Wdr47^{-/-}$ multicilia.** We have recently reported that overexpression of Camsap1 or −3 rescues the polarization defects of $Wdr47$-deficient neurons[15]. To understand whether excessive Camsaps could also functionally compensate for the $Wdr47$

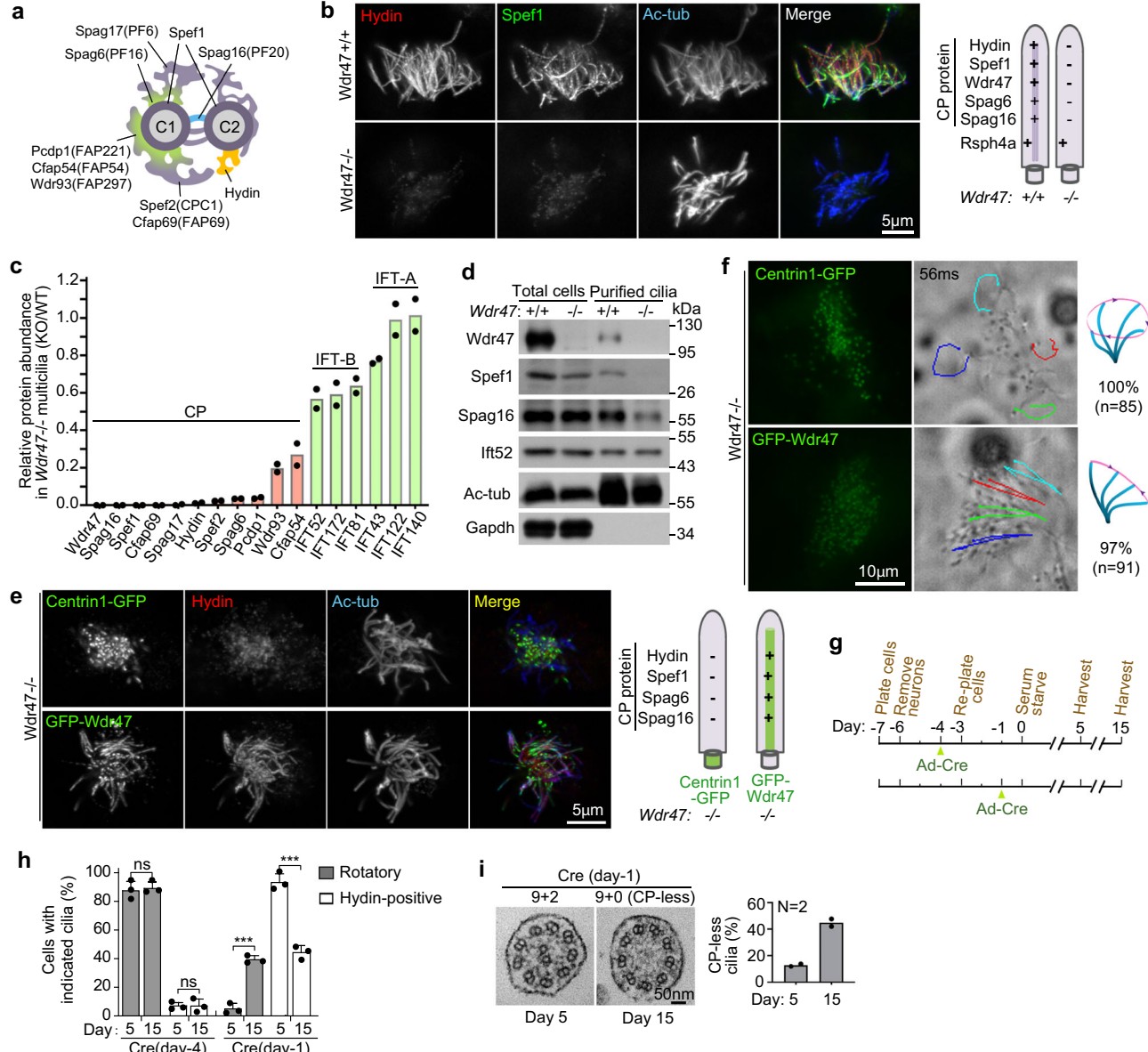

**Fig. 2 Wdr47 is required for both ciliary localization of CP proteins and CP maintenance. a** Diagram for detailed CP locations of the indicated proteins, referring to literature[23,36]. C1 central MT 1, C2 central MT 2. **b** Wdr47 deficiency markedly reduced the immunofluorescence of ciliary CP proteins. Wild-type or Wdr47[−/−] mEPCs fixed at day 10 were immunostained, followed by confocal microscopy. Representative micrographs for Hydin and Spef1 and a summary are shown (n = 2 biological replicates). Ac-tub was used as ciliary marker. Rsph4a, a radial spoke protein, served as negative control. Refer to Supplementary Fig. 3 for the supporting images. **c, d** Wdr47 deficiency markedly reduced ciliary levels of CP proteins. Multicilia purified from day-10 wild-type or Wdr47[−/−] mEPCs were used for label-free quantitative (LFQ) mass spectrometry (**c**) or immunoblotting (**d**). The relative ciliary intensities were from two biologically independent experiments, in which the subunits of intraflagellar transport (IFT) complexes served as negative controls. In the immunoblots (**d**), Ift52 served as loading control, whereas Ac-tub and Gapdh were respectively used as positive and negative controls to indicate the quality of the cilia samples (n = 2 biological replicates). **e** Expression of GFP-Wdr47 in Wdr47[−/−] mEPCs restored ciliary localization of CP proteins. mEPC progenitors from Wdr47[−/−] E18.5 embryos were infected with lentivirus at one day before serum starvation (day −1) to express GFP-Wdr47 or Centrin1-GFP (negative control). At day 10, the cells were fixed for confocal microscopy. Refer to Supplementary Fig. 4 for additional supporting images (n = 3 biological replicates). **f** Expression of GFP-Wdr47 in Wdr47[−/−] mEPCs restored the planar beat of multicilia. Wdr47[−/−] mEPCs infected as in **e** were live imaged at day 10 to monitor cilia motility (n = 4 biological replicates). Trajectories of four cilia during the first 56 ms of imaging are shown for each representative mEPC. The corresponding ciliary beat patterns and statistical results are also illustrated. Please also refer to Supplementary Movie 2. **g** Experimental scheme for CP maintenance assays. mEPC progenitors from Wdr47[flox/flox] neonatal littermates were infected with adenovirus to express Cre and examined at day 5 and day 15. **h, i** The Cre expression from day −1 induced progressive CP disassembly. Multiciliated mEPCs treated as in **g** were live imaged for ciliary motility, followed by immunostaining for Hydin and transmission EM. At least 30 cells in the live imaging (**h**), 50 cells in the confocal imaging (**h**), and 50 ciliary cross-sections in the EM (**i**) were scored in each experiment and condition. Three (**h**) and two (**i**) biologically independent experiments were performed. Refer to Supplementary Fig. 5 for representative confocal images. Quantification results are presented as mean ± s.d. Two-sided Student's t test: ns no significance; ***P < 0.001.

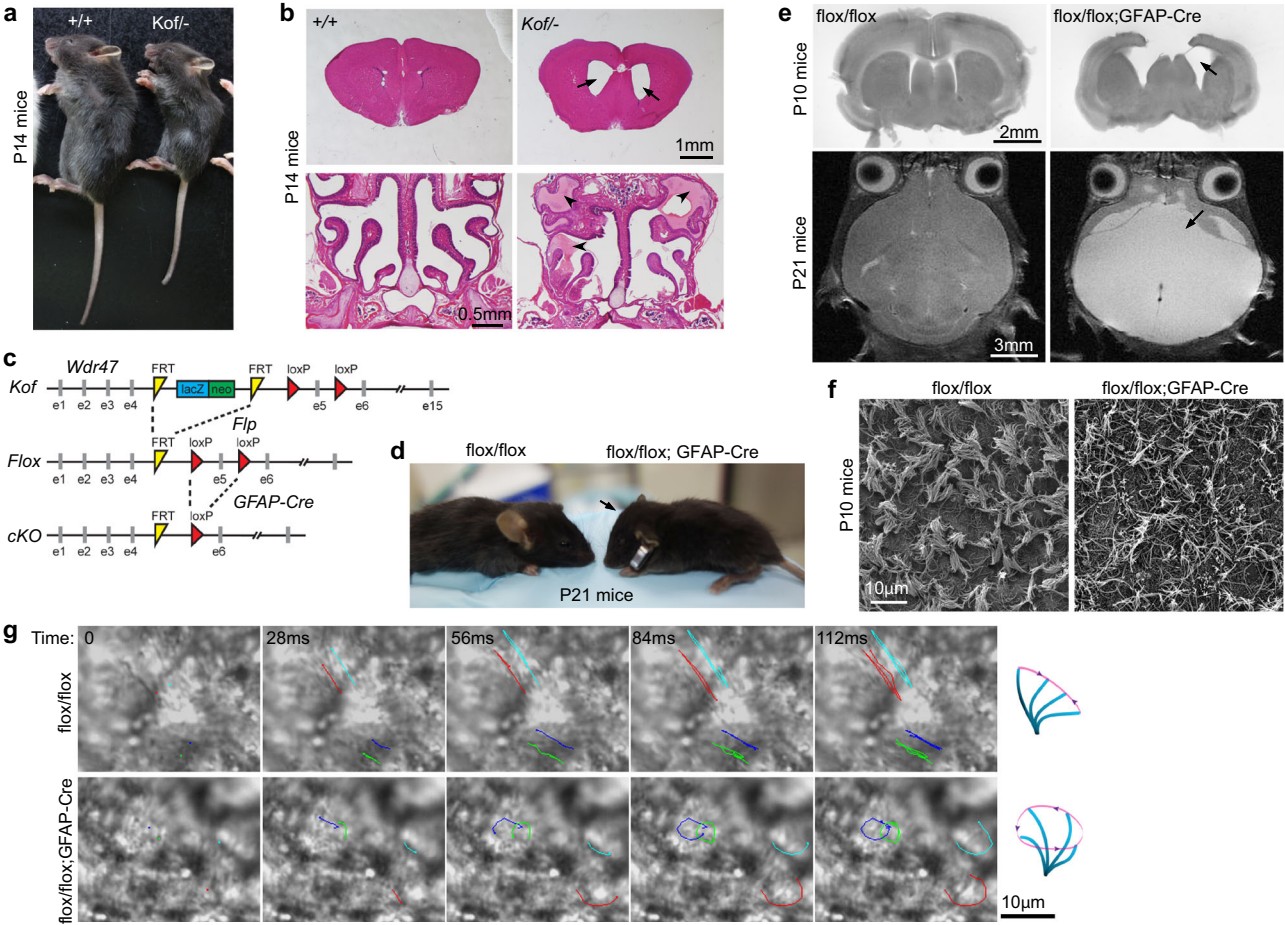

**Fig. 3 Wdr47 deficiency in multicilia results in hydrocephalus and chronic sinusitis. a** Representative littermates of $Wdr47^{Kof/-}$ and wild-type mice.
**b** $Wdr47^{Kof/-}$ mice displayed both hydrocephalus and chronic sinusitis ($n = 3$ biological replicates). Shown are representative HE-stained coronal sections of the brain and the paranasal cavity. Arrows and arrowheads indicate enlarged ventricles and accumulated mucus, respectively. **c** Strategies for generating $Wdr47^{flox/flox}$;GFAP-Cre conditional knockout (cKO) mice to knockout Wdr47 in progenitors of mEPCs. $Wdr47^{Kof/+}$ mice were crossed with Flp mice to produce $Wdr47^{flox/+}$ mice, which were then crossed with GFAP-Cre mice to produce $Wdr47^{flox/+}$;GFAP-Cre mice. **d** The Wdr47 cKO mice displayed a dome-shaped skull (arrow) ($n = 3$ biological replicates). **e** The cKO mice displayed progressive hydrocephalus (arrows) as shown by coronal brain sections at P10 and transverse magnetic resonance images at P21. Experiment was performed once. **f, g** Multicilia in the $Wdr47^{flox/+}$;GFAP-Cre ependyma were disorganized under scanning EM and rotatory in live imaging, acquired at 140 frames per second (fps) using live brain slices of P10 mice. Scanning EM was performed twice. Live imaging was carried out once. Trajectories of four traceable cilia over time are shown on representative image sequences.

deficiency in mEPCs, we overexpressed GFP-tagged Camsaps or Centrin1 (negative control) in $Wdr47^{-/-}$ mEPCs through lentiviral infection (Fig. 5a). $Wdr47^{-/-}$ mEPCs expressing Centrin1-GFP still displayed rotatory multicilia, whereas the majority of the GFP-Camsap1-expressing cells displayed planar ciliary beat (Fig. 5b, c and Supplementary Movie 3). Different cilia, however, tended to beat towards different directions (Fig. 5b and Supplementary Movie 3). The GFP-Camsap1 overexpression also increased the percentage of cells with mixed ciliary beat patterns by 3.1-fold (Fig. 5b, c). Although GFP-Camsap2 and GFP-Camsap3 were expressed in low levels (Fig. 5a), percentages of mEPCs with planar and mixed beat patterns still increased by 6.5-fold and 10.9-fold, respectively, as compared to the Centrin1-GFP-positive cells (Fig. 5b, c). When fixed cells were examined, GFP-Camsaps markedly rescued the ciliary Hydin localization as compared to Centrin1-GFP (Fig. 5d, e). These results suggest a partial rescue of CP by Camsaps overexpression.

Next, we examined the ciliary ultrastructure. As we were unable to recognize GFP-positive cells in EM, we extensively examined the GFP-Camsap1 samples due to their high viral infection efficiency (>80%), exogenous expression levels

(Fig. 5a), and rescue effects on the planar beat pattern and ciliary Hydin localization (Fig. 5b–e). 48.9% of their axonemal cross-sections displayed two central MTs; 3.5% displayed abnormal central MT numbers (1 or 3-to-6) (Fig. 5f, g). Occasionally we observed one dislocated central MT outside the central lumen ($n = 2$), (Fig. 5f) or extra doublet-like MTs ($n = 3$) (Supplementary Fig. 6a). Central MTs were observed to extend from the ciliary base to the tip in longitudinal sections (Fig. 5f). We also observed multiple short MTs at the tip (Fig. 5f), suggesting that the incidence of axonemes with extra central MTs (2.6%; Fig. 5g) is probably underestimated due to quantifications on cross-sections. By contrast, none of the axonemal cross-sections (total $n = 307$) from the Centrin1-GFP samples displayed visible central MT(s). We also examined a set of samples for GFP-Camsap2 and GFP-Camsap3. In all, 27% of axonemal cross-sections from the Camsap3 sample contained either one (1/173) or two (46/173) central MTs (Fig. 5g). All cross-sections from the Camsap2 sample ($n = 239$), however, were CP-less, possibly due to its low expression level and efficacy because EM does not allow choosing cells expressing relative high levels of GFP-Camsap2.

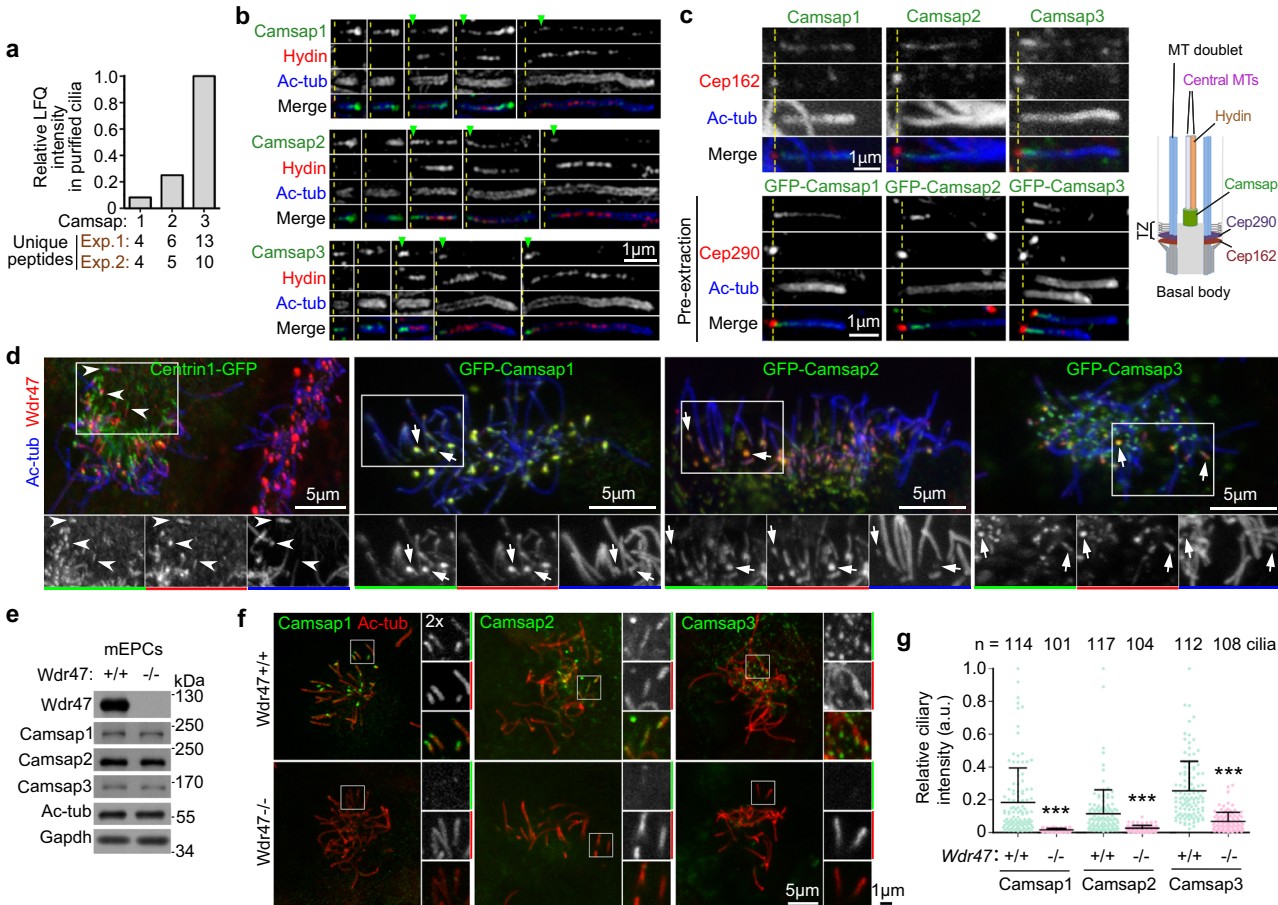

**Fig. 4 Camsaps colocalize with Wdr47 in multicilia. a** Camsaps were detected in purified wild-type multicilia by mass spectrometry. LFQ intensities (relative to that of Camsap3) were from one of the two independent experiments described in Fig. 2c. Unique peptide counts from both experiments are listed. **b** Camsaps localized to ciliary tip and central lumen prior to Hydin (CP marker) in short cilia and preferentially to the bottom region of CP (arrowheads) in long cilia. Ac-tub labels axonemes. Dashed lines mark the base of ciliary shafts based on the immunostaining of Ac-tub. Representative 3D-SIM images of cilia were from mEPCs at day 7. n = 4 biological replicates. **c** Representative cilia showing that Camsaps-decorated CP base regions were above TZ. Cultured mEPCs were immunostained for Ac-tubulin, Cep162, and one of Casmsaps, followed by confocal microscopy (top panels). mEPCs infected with lentiviral particles to express GFP-tagged Camsaps were extracted with 0.5% Triton X-100 for 30 s prior to fixation to remove soluble protein fractions. The samples were then immunostained, followed 3D-SIM (bottom panels). Dashed lines mark the position of TZ based on the immunostaining of Cep162 (top panels) or Cep290 (bottom panels). n = 3 biological replicates. An illustration is provided to aid understanding. **d** Colocalization of ciliary Camsaps with Wdr47. Cultured mEPCs were infected with lentiviral particles at 1 day before serum starvation (day −1) to respectively express GFP-Camsaps and fixed at day 7, followed by immunostaining and confocal imaging. Arrows point to representative short cilia. Centrin1-GFP served as negative control. Arrowheads denote representative cilia that indicate different localization patterns of Wdr47 and Centrin1-GFP. n = 4 biological replicates. **e–g** Wdr47 deficiency impaired the ciliary localization of Camsaps. Cultured Wdr47+/+ or Wdr47−/− mEPCs at day 10 were lysed for immunoblotting (**e**) (n = 3 biological replicates) or fixed for immunostaining (**f**). The framed areas were magnified to show details. Relative intensities (**g**) were obtained by normalizing fluorescent intensities of Camsaps to those of Ac-tub in corresponding cilia. At least 101 cilia were quantified in each condition from three biological independent experiments. The bars and errors represent mean and s.d., respectively. Two-sided Student's t test: ***P < 0.001.

Taken together, we conclude that increasing the total levels of Camsaps partially compensate for Wdr47 deficiency in the central MT formation.

**Wdr47 recruits Camsaps to the central lumen for proper CP formation.** As the partial rescue effects of overexpressed Camsaps (Fig. 5a–g) suggested a role of Wdr47 for efficient production and proper positioning of central MTs, we speculated that Wdr47 might function by increasing the regional ciliary concentration of Camsaps. We have previously shown that Wdr47 interacts with Camsaps through its N-terminal region (WdrN) but not the C-terminal WD40 repeats (WdrC) (Fig. 5h)[15]. When expressed in Wdr47−/− mEPCs, GFP-Wdr47 entered multicilia and rescued the ciliary localizations of endogenous Camsap1 and Hydin, the CP marker (Fig. 5i; also see Fig. 2e). GFP-WdrC, but not GFP-WdrN,

strongly localized into multicilia (Fig. 5i). Neither construct, however, was able to function as the full-length to restore ciliary Camsap1 and Hydin (Fig. 5i) or the planar ciliary beat (Fig. 5j vs. Fig. 2f). Wdr47 thus requires both its Camsap-interacting and cilia-localization regions for the ciliary enrichment of endogenous Camsaps and, consequently, proper CP formation.

We further examined the detailed localization of GFP-WdrC through super-resolution microscopy and found that GFP-WdrC was distributed in the ciliary central lumen (Fig. 5k). As Wdr47−/− multicilia are CP-less (Fig. 1i), such a result suggests that this C-terminal region can target Wdr47 to the central lumen independently of central MTs. Taken together, we conclude that Wdr47 binds to Camsaps through its N-terminal region and targets them to the ciliary central lumen through its C-terminal region for efficient and proper CP formation.

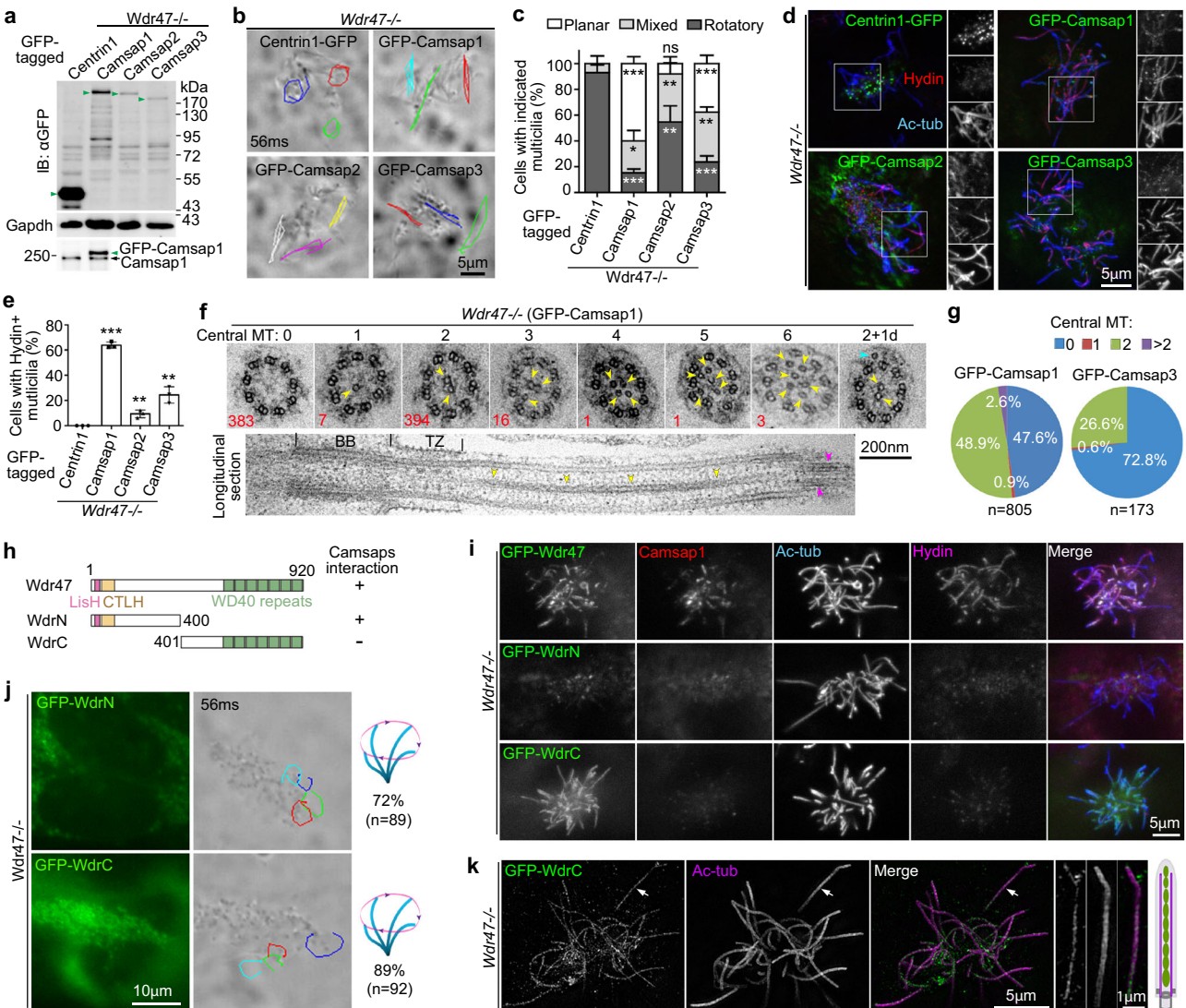

**Fig. 5 Wdr47 concentrates Camsaps to the central lumen for efficient and proper central MT formation. a** Expression levels of exogenous Camsaps and Centrin1 (arrowheads) in *Wdr47*^-/- mEPCs. *Wdr47*^-/- mEPCs were infected with lentivirus at day −1, day 2, and day 5 to express GFP-tagged Camsaps or Centrin1 (negative control). The cells were harvested at day 10 for immunoblotting (IB) or assays in **b**–**g**. >80% of the cells were usually GFP-positive. The average GFP-Camsap1 level was estimated to be 2.3-fold over that of endogenous Camsap1 when IB was performed using anti-Casmsap1 antibody (bottom panel). Gapdh served as loading control. $n = 3$ biological replicates. **b, c** Effects on ciliary beat pattern. Trajectories of three cilia during the first 56 ms of live imaging are shown for each mEPC (**b**). Please refer to Supplementary Movie 3. Quantification results (**c**) were from three biologically independent experiments. At least 40 cells were scored in each experiment and condition. Data are presented as mean ± s.d. Two-sided Student's *t* test: *$P < 0.05$; **$P < 0.01$; ***$P < 0.001$. **d, e** Effects on ciliary localization of Hydin. Separate grayscale channels are shown for the framed regions in the confocal images (**d**). Quantification results (**e**) were from three biologically independent experiments. At least 66 cells were scored in each experiment and condition. Data are presented as mean ± s.d. Two-sided Student's *t* test against the Centrin1-GFP populations: **$P < 0.01$; ***$P < 0.001$. **f** Representative axonemal ultrastructure. Arrowheads point to central MTs. The cyan arrowhead in the last cross-section indicates a dislocated MT outside the central lumen (marked with a postfix "d" in the central MT numbers), whereas the magenta arrowheads in the longitudinal section point to two possible pairs of short central MTs at the ciliary tip. The total count for each type (red number) was from three biologically independent experiments. See Supplementary Fig. 6a for additional examples. **g** Pie charts summarizing the percentages of axoneme sections with different central MT numbers. Quantification results of GFP-Camsap1 and GFP-Camsap3 were from three and one biologically independent experiments, respectively. **h** Diagrams of Wdr47 and its deletion constructs. Their abilities to interact with Camsaps are referred to our previous publication[15]. **i, j** Both the N- and C-termini of Wdr47 were required for ciliary Camsap1 enrichment and the CP formation. *Wdr47*^-/- mEPCs infected with lentivirus at day −1 to express the GFP-tagged proteins were fixed (**i**) or live imaged (**j**) at day 10. Hydin served as CP marker in **i** ($n = 2$ biological replicates). Trajectories of four cilia during the first 56 ms of live imaging are shown for each representative mEPC in **j**. Quantification results on the rotatory ciliary beat pattern are also provided ($n = 4$ biological replicates). **k** WdrC localized into the central lumen independently of CP. *Wdr47*^-/- mEPCs expressing GFP-WdrC were imaged by 3D-SIM ($n = 3$ biological replicates). The arrow-indicated cilium was magnified 2-fold to show central-lumen localization of GFP-WdrC as depicted by the diagram.

**Camsaps promote the central MT formation by stabilizing the MT minus end**. We have previously shown that Camsap3 can recruit Wdr47 to MT minus ends[15], implying that Wdr47 might target Camsaps by binding to regions outside their MT-binding domains. To clarify this, we mapped the Wdr47-interacting region of Camsap1 by co-immunoprecipitation. Indeed, the linker region between the CH and the first coiled coil (CC) domains of Camsap1 (312-858 aa) strongly interacted with Wdr47, whereas the CC-containing region and the CKK domain involved in the MT association[10–13] were dispensable (Fig. 6a, b).

Camsaps function in MT dynamics by stabilizing the minus end of pre-existing MTs[10,12,13]. It is known that, although the CKK domain alone is sufficient for binding to the MT minus end, a longer fragment containing the CC region displays markedly increased lattice-binding and minus end-stabilization ability[10–13]. To understand whether Camsaps used the similar mechanism to induce central MT formation, we overexpressed constructs containing the CKK domain and a longer CC-containing fragment (LC) as GFP fusion proteins in $Wdr47^{-/-}$ mEPCs (Fig. 6c, d). Both types of constructs displayed significant rescue effects on ciliary beat patterns as compared to Centrin1-GFP (Fig. 6e and Supplementary Movie 4), suggesting that the CKK domain alone is already functional. Consistently, all the constructs displayed ciliary localizations as well as an ability to rescue the ciliary localization of Hydin in the $Wdr47^{-/-}$ mEPCs (Fig. 6f and Supplementary Fig. 7). The LC constructs generally yielded higher percentages of ciliary Hydin-positive mEPCs than the CKK constructs (Fig. 6f and Supplementary Fig. 7).

EM further confirmed that all the constructs of Camsaps induced central MT formation in the $Wdr47^{-/-}$ multicilia (Fig. 6g, h). Notably, 70% of the axonemal cross-sections from the GFP-Ca1LC samples contained one or more central MTs and 21% contained >2 (up to 13) central MTs (Fig. 6g, h). Furthermore, 28 sections (8%) contained dislocated central MTs outside the central lumen (Fig. 6h). By contrast, 25% of the cross-sections from the GFP-Ca1CKK samples contained central MTs (Fig. 6g). Among them, three sections (2%) contained dislocated central MT(s). In addition, axonemes with an extra MT doublet were also observed for both constructs (Supplementary Fig. 6b, c). We observed both long and short central MTs in longitudinal axonemal sections (Supplementary Fig. 6d). In comparison, 38% and 44% of the cross-sections respectively from the GFP-Ca2LC and GFP-Ca3LC samples contained one or more central MTs, with 3% and 18% of the total cross-sections containing >2 (up to 10) central MTs (Fig. 6g, h). The incidences of central MT-containing cross-sections from the GFP-Ca2CKK and GFP-Ca3CKK samples were 15% and 38%, respectively (Fig. 6g).

These results further strongly support a direct role of Camsaps in the CP formation by binding and stabilizing the minus ends of central MTs. The prominent rescue effects of GFP-Ca2LC and GFP-Ca2CKK (Fig. 6e–g), despite their low expression levels relative to the corresponding constructs of Camsap1 and Camsap3 (Fig. 6d), also confirm that Camsap2 is involved in the CP formation. Furthermore, as Camsaps are unable to nucleate MTs[10–13], the formation of extra and even 13 central MTs indicates that the available central MT seeds in an axoneme can largely exceed two.

**Central MT formation requires the MT-severing activity of Katanin**. The seeds for non-centrosomal MTs usually come from severed MTs or MTs released from the γ-tubulin ring complex[5,6,12,13]. As the MT-severing enzyme Katanin directly binds to Camsap2 and −3[12,40] and protozoan Katanin is essential for the CP formation and has been shown to bind to and severing

axonemal MT doublets[61–63], we speculated that Katanin might function to generate the central MT seeds for mammalian multicilia. Immunostaining revealed that Katanin p60 displayed multiciliary localizations in mEPCs, abundant at the tip of short cilia (Fig. 7a). GFP-p60 also displayed similar ciliary localizations and co-localized with Camsaps at the ciliary tip, when Camsap1 was used as marker (Fig. 7b).

To investigate whether the MT severing activity of Katanin is required for the central MT formation, we overexpressed GFP-p60 or GFP-p60$^{K257A}$, a point mutant acting as a dominant inhibitor[64], in wild-type mEPCs (Fig. 7c). Like p60, the dominant mutant p60$^{K257A}$ also entered multicilia (Fig. 7d). Wild-type mEPCs overexpressing Centrin1-GFP rarely had Hydin-negative cilia (0.9%; $n = 117$) (Fig. 7d). The GFP-p60 overexpression moderately increased the incidence (8.9%; $n = 90$). By contrast, multicilia in 46.7% of mEPCs positive for GFP-p60$^{K257A}$ ($n = 105$) lacked Hydin (Fig. 7d). Consistently, the percentage of mEPCs with rotatory ciliary beat pattern increased moderately (5.3-fold) in the GFP-p60 populations and markedly (40.7-fold) in the GFP-p60$^{K257A}$ populations as compared to the Centrin1-GFP populations (Fig. 7e and Supplementary Movie 5). EM analyses revealed increased incidences of CP-less cross-sections (2.3-fold in the GFP-p60 samples and 9.3-fold in the GFP-p60$^{K257A}$ samples) as well (Fig. 7f). Notably, cross-sections containing 1 (3%) and >2 (4%) central MTs emerged in the GFP-p60 samples, which were not observed in either the Centrin1-GFP or the GFP-p60$^{K257A}$ samples (Fig. 7f). The MT-severing activity of Katanin is thus critical for proper mammalian CP formation.

Next we explored whether the rescue effect of Camsaps in $Wdr47^{-/-}$ mEPCs required Katanin activity by co-expressing RFP-p60$^{K257A}$ or RFP-p60. We initially performed the experiments with GFP-Camsap1 because it displayed the strongest rescue effect (Fig. 5a–g). Moreover, as Camsap1 does not interact with Katanin[1], its overexpression would not influence the effects of RPF-p60 and RFP-p60$^{K257A}$. Compared to Centrin1-RFP, co-expression of RFP-p60$^{K257A}$ with GFP-Camsap1 induced a 3.3-fold decrease in percentage of cells with planar ciliary beat and a 2.8-fold increase in that with rotatory ciliary beat (Fig. 7g and Supplementary Movie 6). In comparison, co-expression of RFP-p60 did not significantly alter the rescue effects of GFP-Camsap1 in $Wdr47$-deficient mEPCs (Fig. 7g and Supplementary Movie 6). We then examined GFP-Ca3LC, the Camsap3 deletion mutant displaying a remarkable rescue effect (Fig. 6d–h) and lacking Katanin-binding region (Fig. 6c)[40]. Compared to Centrin1-RFP, co-expression of RFP-p60$^{K257A}$ also similarly compromised the rescue efficiency of GFP-Ca3LC (Fig. 7g). Therefore, Katanin activity is important for the Camsap overexpression-induced central MT formation.

Therefore, like protozoan Katanin[61–63], mammalian Katanin also functions in the CP formation. As tubulin dimers can be transported into cilia through intraflagellar transport[35], we modified the model by Sharma and colleagues[63] and propose that Katanin provides central MT seeds by severing the tip region of axonemal MT doublets (Fig. 7h).

## Discussion

Our results indicate that Wdr47 ensures the efficient formation and accurate positioning of central MTs by concentrating Camsaps into cilia. Wdr47 is a prerequisite because CP completely failed to form upon $Wdr47$ deficiency (Figs. 1 and 2, and Supplementary Figs. 2 and 3). Ciliary Camsaps co-localized with Wdr47 mainly at the tip of nascent short cilia and the base of central lumen in long cilia and became markedly reduced in $Wdr47$-deficient mEPCs (Figs. 1d, e and 4b–g). Although central MTs can form in $Wdr47^{-/-}$ mEPC when the total levels of Camsaps were increased through

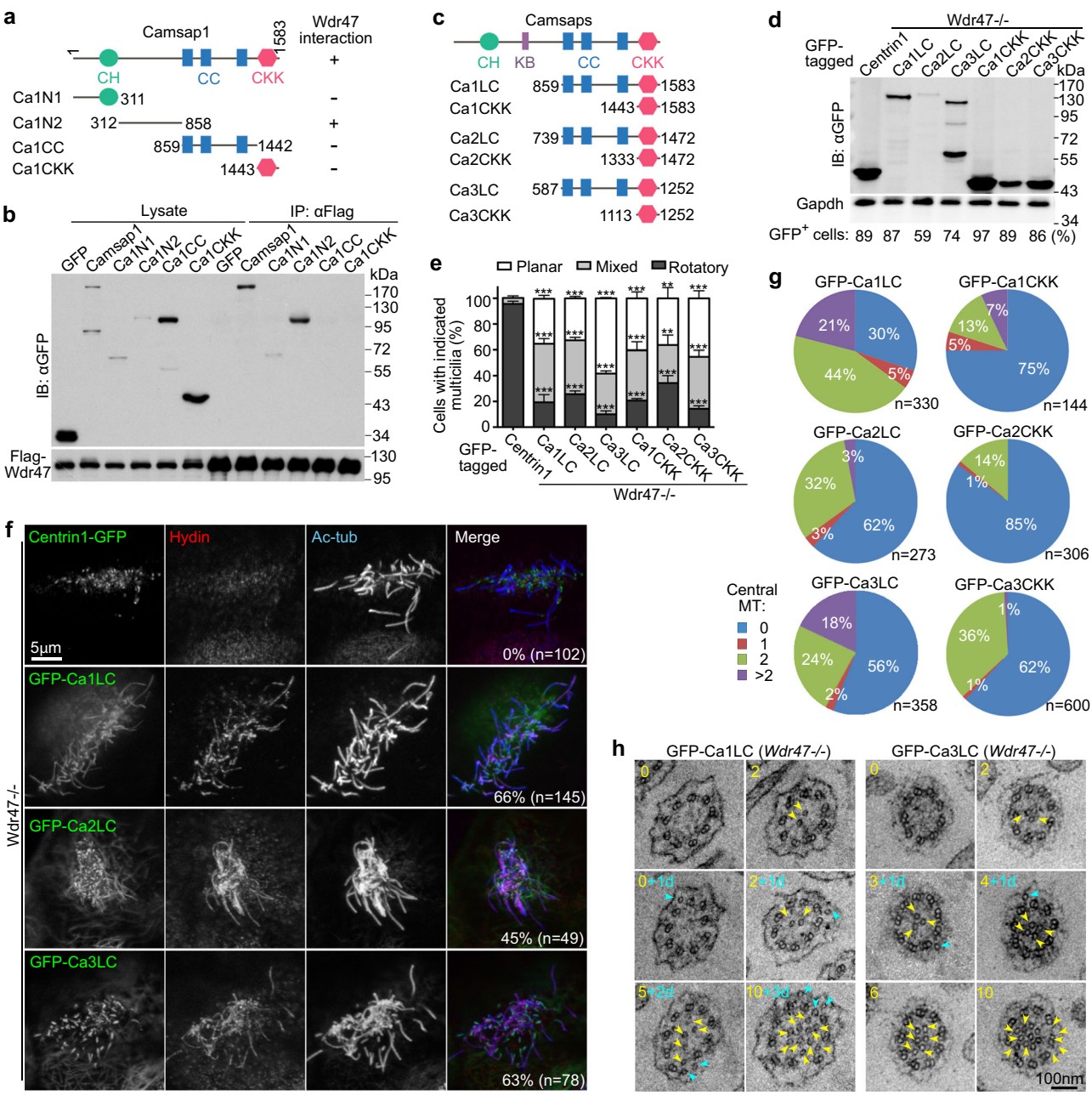

**Fig. 6 Camsaps induce central MTs by stabilizing pre-existing MT seeds. a**, **b** Mapping the Wdr47-binding region of Camsap1. GFP-tagged Camsap1 and constructs (**a**) were co-expressed with Flag-Wdr47 in HEK293T cells, followed by co-immunoprecipitation using anti-Flag resin and immunoblotting (**b**) ($n = 3$ biological replicates). Structural regions in **a**: CH Calponin homology, CC coiled coil, CKK MT minus end-binding. **c** Diagrams of Camsap deletion constructs for experiments in **d**–**h**. KB Katanin-binding domain, which appears to exist only in Camsap2 and -3[40]. **d** Expression levels of Camsap deletion constructs. $Wdr47^{-/-}$ mEPCs were infected with lentivirus at day −1, day 2, and day 5 to overexpress GFP-tagged Camsap constructs or Centrin1-GFP (negative control) and harvested at day 10 for immunoblotting (**d**) and experiments in **e**–**h**. Percentages of GFP-positive cells for a typical set of the cells were indicated. At least 134 multiciliated cells were scored for each example ($n = 2$ biological replicates). **e** Effects on ciliary beat patterns, quantified from three biologically independent experiments and presented as mean ± s.d. At least 36 multiciliated cells expressing Centrin1-GFP and 80 multiciliated cells expressing the Camsap constructs were scored in each time and condition. Two-sided Student's *t* test: ***$P < 0.001$. Please refer to Supplementary Movie 4. **f** Typical confocal micrographs of multiciliated cells expressing the LC constructs or Centrin1-GFP. Hydin served as CP marker. Percentages of GFP and ciliary Hydin double-positive cells are shown. See Supplementary Fig. 7 for micrographs of the CKK constructs ($n = 3$ biological replicates). **g**, **h** Pie charts (**g**) and representative axoneme cross-sections (**h**) for the status of central MTs. Numbers of central MTs (arrowheads) are marked in **h**, with the postfix "d" standing for dislocated central MTs outside the central lumen (cyan arrowheads). Experiments were performed once. See Supplementary Fig. 6b–d for additional axoneme examples.

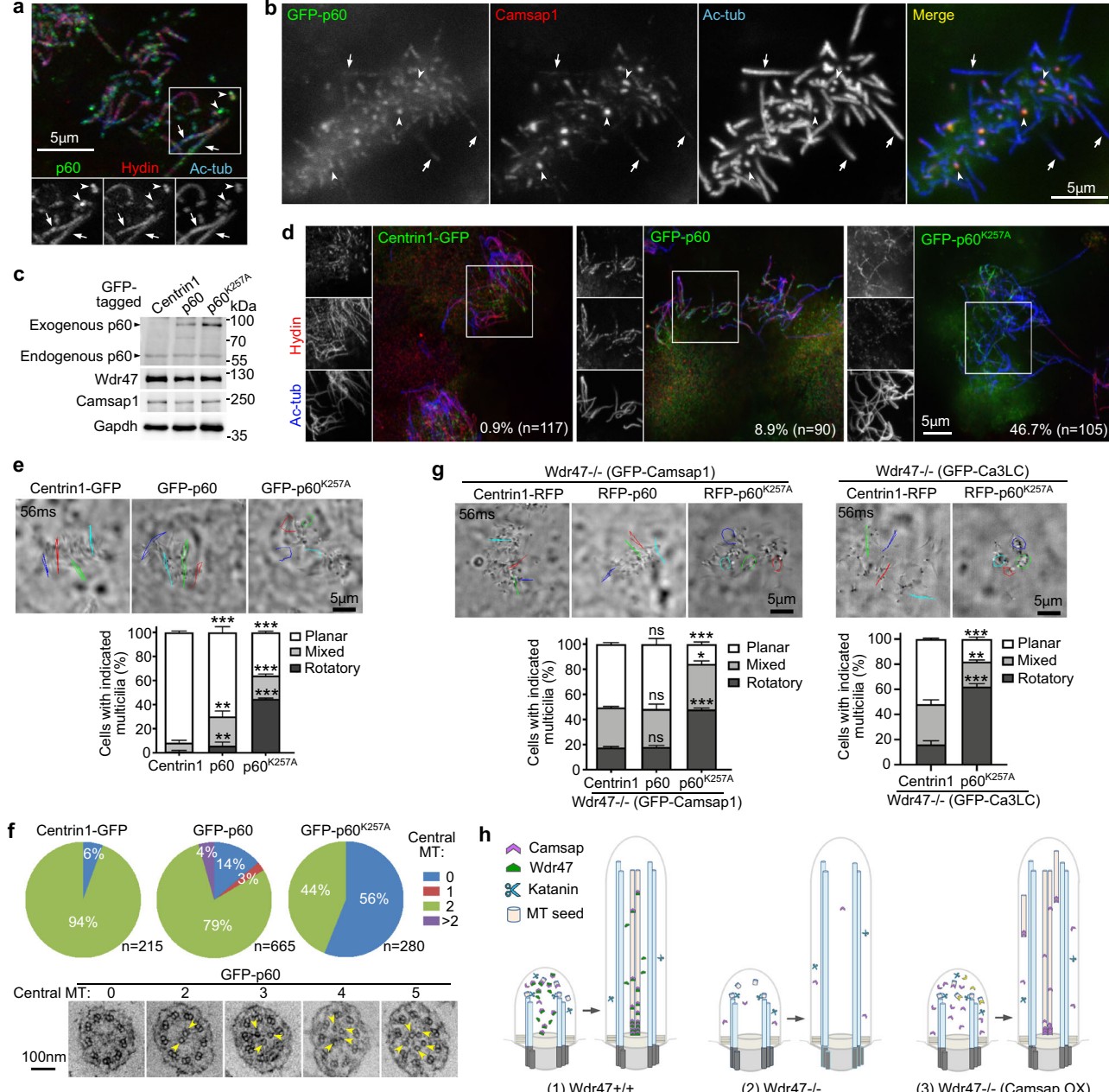

overexpression (Figs. 5a–g and 6, and Supplementary Fig. 7), they emerged with low efficiency and abnormal numbers, and sometimes at a wrong location: comparing to a near 100% efficiency of the CP formation in wild-type ependymal and airway multiciliated cells (Fig. 1i and Supplementary Fig. 2)[28,30,65], only <50% of ependymal ciliary cross-sections were rescued to two central MTs and at least 30% were still CP-less (Figs. 5f, g and 6g, h). Furthermore, up to 21% contained more than two central MTs, and dislocated singlet MT(s) outside the central lumen were also observed (Figs. 5f, g and 6g, h, and Supplementary Fig. 6d). As Camsaps do not induce MT nucleation[10–13], the emergence of excessive numbers (up to 13 in our EM results) of central MTs (Figs. 5f, g and 6g, h) indicate that Camsaps function by stabilizing the minus ends of pre-existing MT seeds whose number can largely exceed two in a cilium. Our interaction and localization analyses (Figs. 5i, k and 6a, b) further suggest that Wdr47 binds to the N2 region of Camsaps through its N-terminal region and recruits them to the ciliary central lumen through its C-terminal region.

Therefore, the Wdr47-Camsap interaction allows Camsaps to function at low physiological concentrations to avoid production of extra central MTs. How a cilium manages to produce precisely two central MTs, however, still requires future quantitative studies.

Camsap1-3 appear to function redundantly and contribute collectively to the central MT formation. Different Camsaps share common and also display individual properties[10–14]. As mEPCs express all three proteins concomitantly (Fig. 4), the contribution of each Camsap may be affected by both expression level and efficacy relative to the others (Figs. 5 and 6). Although multicilia-related defects have not been documented for Camsap1- or Camsap2-deficient mice[14,18], during the revision process we noticed that a recent publication reports that Camsap3 localizes to the base of axonemes in murine nasal multiciliated cells and Camsap3 deficiency results in the CP loss in 77.4% axonemal cross-sections of the cells[66]. Consistently, Camsap3-deficient mice display severe nasal airway blockage[66], similar to our Wdr47[Kof/-] mice (Fig. 3b). In sharp contrast to the severe hydrocephalus of

**Fig. 7 Katanin provides central MT seeds by severing axonemal MTs.** Pooled data (mean ± s.d.) were from three biologically independent experiments. Student's t-test, ns, no significance; **P < 0.01; ***P < 0.001. **a, b** Katanin p60 displayed multiciliary localizations. Wild-type mEPCs were fixed at day 7 either directly for visualizing endogenous p60 through immunostaining (**a**) or after infection with lentivirus at day −1 to express GFP-p60 (**b**). Ac-tub and Hydin served as ciliary and CP markers, respectively. Arrowheads and arrows point to representative short cilia and long cilia, respectively. n = 3 biological replicates. **c** Expression level of GFP-p60 and GFP-p60$^{K257A}$ in mEPCs. Wild-type mEPCs were infected with lentivirus at day −1 and day 2 to express Centrin1-GFP, GFP-p60 or GFP-p60$^{K257A}$ and harvested at day 10 for assays in **c–f**. Gapdh served as loading control in immunoblotting (**c**). Note that the expression of GFP-tagged p60 or p60$^{K257A}$ had little influence on the levels of endogenous p60, Wdr47, and Camsap1, as compared to Centrin1-GFP. **d** p60$^{K257A}$ abolished the Hydin localization in multicilia. The framed regions of confocal micrographs are shown in separate grayscale channels. Percentages of GFP-positive and ciliary Hydin-negative cells are shown. n = 3 biological replicates. **e** Effects of GFP-p60 or GFP-p60$^{K257A}$ overexpression on ciliary beat patterns. Trajectories of four cilia during the first 56 ms of imaging are shown for each EPC. Please refer to Supplementary Movie 5. At least 34 cells were scored in each experiment and condition. Three biologically independent experiments were performed. Error bars present as mean ± s.d. Two-sided Student's t test: **P < 0.01; ***P < 0.001. **f** Effects of GFP-p60 or GFP-p60$^{K257A}$ overexpression on the central MT formation. Representative cross-sections in the GFP-p60 samples are presented. Arrowheads point to central MTs. n = 3 biological replicates. **g** RFP-p60$^{K257A}$ attenuated the rescue effects of Camsaps overexpression on ciliary beat patterns in Wdr47$^{−/−}$ mEPCs. Wdr47$^{−/−}$ mEPCs were infected with lentivirus at day −1, day 2, and day 5 to co-express GFP-Camsap1 or GFP-Ca3LC with the indicated RFP-tagged proteins and live imaged at day 10. Trajectories of four cilia during the first 56 ms of imaging are shown for each representative EPC. At least 45 GFP and RFP double-positive cells in the GFP-Camsap1 group and 31 double-positive cells in the GFP-Ca3LC group were scored in each experiment and condition. Three biologically independent experiments were performed. Error bars present as mean ± s.d. Two-sided Student's t test: *P < 0.05; **P < 0.01; ***P < 0.001. Please also refer to Supplementary Movie 6. **h** A summarizing model for central MT formation: (1) central MT seeds are generated by Katanin from peripheral MTs in short multicilia, stabilized at their minus ends by Camsaps, and recruited to the central lumen through Wdr47; (2) Wdr47 deficiency abolishes the CP formation by markedly decreasing ciliary Camsaps; (3) Increasing total Camsap levels through overexpression induces central MT formation but with low efficiency and various abnormalities. See Discussion section for details.

---

our Wdr47$^{flox/flox}$;GFAP-Cre cKO mice (Fig. 3c–e), Camsap3-deficient mice display only mild or no hydrocephalus phenotypes[66]. Two subsequent publications report that Camsap3-deficient mice have normal beating cilia in ependymal cells[67] and normal CP formation in oviduct multicilia[68]. These results also echo our proposed redundant and collective effects of Camsaps. Future systematic analyses using single, double, and triple gene knockout mice of Camsaps will be still required to determine detailed contributions of individual Camsaps and their collective effects on ciliary CP formation of different tissue cells.

Our results suggest that Katanin is involved in the production of central MT seeds. p60 and p80 subunits of Katanin are in fact the only two proteins known to be essential to the CP formation in protozoa[48,61–63]. Chlamydomonas p80 specifically distributes in ciliary outer doublet compartment[61], whereas overexpressed GFP-p60 has been shown to specifically bind to and sever outer doublets in Tetrahymena[63]. Protozoan Katanin has also been proposed to provide central MT seeds by severing MTs nucleated from intraflagellar γ-tubulin[62] or free tubulin dimers for CP assembly by severing outer doublets[63]. We found that both endogenous p60 and GFP-p60 display ciliary localization and also enrich at the tip of short cilia (Fig. 7a, b). Katanin activity is also important for Camsaps-mediated CP formation (Fig. 7e–g). As overexpression of GFP-p60 induced CP loss in a portion of ependymal axonemes (Fig. 7e, f), mammalian Katanin unlikely functions by providing tubulin dimers for the CP formation. On the other hand, no evidence to date suggests a localization of γ-tubulin in mammalian multicilia[69–71]. As the observations of up to 13 central MTs in our rescue experiments (Figs. 5f, g and 6g, h) provide solid evidence for the presence of excessive central MT seeds and nascent central MTs have been shown to emerge initially from the top region of Chlamydomonas flagella[46], we modified the previous models and propose that mammalian Katanin severs the tip of axonemal outer MTs to generate central MT seeds for Camsaps to bind and stabilize at early stages of ciliogenesis (Fig. 7h). Consistently, we observed doublet-like MTs additional to the nine outer doublets upon the overexpression of Camsap1 or its deletion constructs (Supplementary Fig. 6a–c). Future studies will be required to verify whether Katanin p60/p80 enters multicilia by binding to Camsap2 and Camsap3[12,40]. In addition, two paralogues of Katanin p60, Katnal1 and Katnal2,

also affect the growth and motility of multicilia[38,72,73]. Katnal1 has also been shown to sever MTs to build the dense MT arrays in Drosophila mechanosensory cilia[38,72–74]. As the overexpress of p60$^{K257A}$ did not completely abolish the CP formation in mEPCs (Fig. 7e, f), whether Katnal1 and Katnal2 have a role in metazoan central MT seeds production also needs to be clarified in the future.

Taken together, we propose a model that Wdr47, Camsaps, and Katanin function together to produce the CP of mammalian multicilia: Katanin severs the tip of outer MTs to generate central MT seeds in nascent short cilia, Camsaps stabilize the seeds by binding to their minus ends, and Wdr47 concentrates Camsaps and facilitates their targeting into ciliary central lumen so that the Camsaps-bound MT seeds eventually develop into CP following the elongation of cilia (Fig. 7h). In addition to the CP formation, emerging clues of their corporative actions are reported in neurons[15,16,18,41,42,75] and epithelial cells[17,19,76]. Other non-centrosomal MT arrays requiring Camsaps[5–7] could involve Katanin and Wdr47 as well. Both Katanin and Wdr47 are expected to bind between the CH domain and the first CC region of Camsap2 and -3 (Fig. 6a, b)[40]. Therefore, these three groups of proteins might similarly constitute a team for the formation of non-centrosomal MT arrays in polarized subcellular compartments. It will thus be interesting to clarify their detailed interplays and consequences of the interactions as well in other types of cells.

## Methods

**Plasmids.** The full-length or partial cDNAs for mouse Wdr47 (NM_181400), mouse Camsap1 (XM_006497897), mouse Camsap2 (NM_001347109), and mouse Katanin p60 (NM_011835) were amplified by PCR from total cDNAs from mouse testis, brain or mTECs. The full-length of mouse Camsap3 (NM_027171) was amplified by PCR from the GFP-Nezha plasmid (kindly provided by Dr. Wenxiang Meng, Institute of Genetics and Developmental Biology, Chinese Academy of Sciences)[77]. To express GFP- or RFP- fusion proteins, the cDNA fragments were constructed into the lentiviral expression vector, pLV-EGFP-C1 or pLV-RFP-C1[78], respectively. The cDNAs of WdrN (1-400 aa) and WdrC (401-920 aa) were PCR amplified from the pLV-EGFP-Wdr47 plasmid and constructed into pLV-EGFP-C1. The cDNAs of Camsaps: Ca1N1 (1-311 aa), Ca1N2 (312-858 aa), Ca1CC (859-1442 aa), Ca1CKK (1443-1583 aa), Ca1LC (859-1583 aa), Ca2CKK (1333-1472 aa), Ca2LC (739-1472 aa), Ca3CKK (1113-1252 aa), and Ca3LC (587-1252 aa) were PCR amplified from the pLV-EGFP-Camsaps plasmids and constructed into pLV-EGFP-C1. The K257A mutant of Katanin p60 (900 AAG→GCG) were generated by PCR[64]. For expression of FLAG-fusion proteins, the cDNA fragments were

subcloned into the pcDNA3.1-NFLAG vector. The full-length cDNAs of mouse Wdr47 or mouse Camsap1 (1073-1382 aa) were PCR amplified and constructed into pET28a to express His-tagged antigens for antibody production and into pGEX4T-1 to express GST-fusion proteins for antibody purification. All the primers were list in the Supplementary Table 1. All the constructs were verified by sequencing.

**Mice**. Mice experiments were performed in accordance with the ethical guidelines of Shanghai Institute of Biochemistry and Cell Biology, Chinese Academy of Sciences, and approved by the Institutional Animal Care and Use Committee.

*Wdr47$^{Kof/+}$* mice (*Wdr47$^{tm1a(EUCOMM)Wtsi}$*) were purchased from Wellcome Trust Sanger Institute[79]. *Wdr47$^{+/-}$* and *Wdr47$^{flox/+}$* mice were generated by crossing the *Wdr47 $^{Kof/+}$* mice with *Ella-Cre* or *FLP* mice (Model Animal Research Center of Nanjing University, China), respectively. To knockout *Wdr47* in the GFAP-positive glial cells, Wdr47 *$^{flox/flox}$* mice were crossed with *GFAP-Cre* mice (a gift from Dr. Leping Cheng, Institute of Neuroscience, Chinese Academy of Sciences). Primers used for genotyping are listed in Supplementary Table 1.

**Cell culture, transfection, viral infection, and cilia purification**. Cells were maintained at 37 °C in an atmosphere containing 5% $CO_2$. Unless otherwise indicated, the culture medium was Dulbecco's Modified Eagle's medium (DMEM) supplemented with 10% fetal bovine serum (Ausbian, VS500T), 0.3 mg/ml glutamine (Sigma, G8540), 100 U/ml penicillin (Solarbio P8420), and 100 U/ml streptomycin (Solarbio S8290).

mTECs were isolated and cultured as described previously[78,80]. mTECs were isolated from 4-week C57BL/6J mice. After dissecting the adhered muscle tissue, tracheas were sliced lengthwise and digested in Ham's F-12K medium with 0.15% Pronase E (Sigma, P6911) and 0.1 mg/ml DNase I (Sigma, D5025) overnight at 4 °C. Cells were collected by centrifugation for 5 min at 400 × g at room temperature (r.t.) and resuspended with mTEC basic medium [DMEM-Ham's F-12 medium (Thermo Fisher, 11330-032) supplemented with 3.6 mM sodium bicarbonate, 4 mM L-glutamine, 1% penicillin/streptomycin, 0.25 µg/ml fungizone] with 10% FBS. Cells were plated and incubated at 37 °C for 4 h to allow fibroblasts to attach. mTECs were collected by centrifugation at 400 × g for 5 min, resuspended in mTEC plus medium [mTEC basic medium supplemented with 10 µg/ml insulin (Sigma, I6634), 5 µg/ml transferrin (Sigma, T8158), 0.1 µg/ml Cholera toxin (Sigma, C8052), 25 ng/ml epidermal growth factor (Sigma, E4127), 30 µg/ml bovine pituitary extract (Sigma, P1167), 5% FBS, and 0.05 µM retinoic acid (freshly added; Sigma, R2625)], and seeded into collagen (Sigma, C8897)-coated 6.5-mm Transwells with 0.4-µm-pore polyester membrane insert (Corning, 3470). When cells reached full confluency, air-liquid interface (ALI) was created by removing medium in the upper compartment and replacing medium in the bottom compartment with the mTEC differentiation medium [mTEC basic medium supplemented with 2% Nu Serum (BD, 355100) and 0.05 µM retinoic acid (freshly added)] to induce differentiation. DAPT (Sigma, D5942) was added to 10 µM at day 1 post ALI to increase multiciliated cell differentiation efficiency.

Multiciliated mEPCs were obtained and cultured as described[36,53,81]. P0 C57BL/6J mice telencephala were dissected after removing the cerebellum, olfactory bulbs, and hippocampus with sharp tweezers (Dumont, 1214Y84) in cold dissection solution (161 mM NaCl, 5 mM KCl, 1 mM MgSO$_4$, 3.7 mM CaCl$_2$, 5 mM Hepes, and 5.5 mM Glucose, pH 7.4) under a stereo microscope. The telencephala were digested with 1 ml of the dissection solution containing 10 U/ml papain (Worthington, LS003126), 0.2 mg/ml L-Cysteine, 0.5 mM EDTA, 1 mM CaCl$_2$, 1.5 mM NaOH, and 0.15% DNase I (Sigma, D5025) for 30 min at 37 °C. Cells were dissociated mechanically by pipetting up and down 10 times with a 5-ml pipette and collected by centrifugation at 400 × g for 5 min at r.t. Cells were resuspended with DMEM medium supplemented with 10% fetal bovine serum (FBS) and 1% penicillin/streptomycin, and inoculated into the laminin-coated flask (Sigma, L2020). Neurons were shaken off and removed after culturing for 2 days after inoculation. The remaining cells were further cultured to ~80% confluency (usually 3–4 days) and then transferred into the wells of laminin-coated 29-mm glass-bottom dishes (Cellvis, D29-14-1.5-N) for motility assay or immunofluorescence staining or 75-cm² laminin-coated flasks for cilia purification. After cells were confluent, FBS was removed from the medium to initiate differentiation.

To express exogenous proteins, HEK 293T were transfected with polyethlenimine (PEI, Polysciences, 23966-2), respectively, for 48 h. Lentiviral production and infection were performed as described previously[78]. Cultured mEPCs were infected at day one before serum starvation (day −1) unless otherwise stated. For rescue experiments and expressing GFP-tagged Camsaps, and their deletion constructs in *Wdr47$^{-/-}$* ependymal cells, mEPCs derived from *Wdr47$^{-/-}$* mice were infected with lentivirus at day −1, day 2, and day 5 and assayed at day 10.

Purification of ependymal cilia and LFQ mass spectrometric analyses were carried out as described[53]. To purify ependymal cilia, two 75-cm² flasks of ependymal cells derived from 10 E18.5 *Wdr47$^{+/+}$* or *Wdr47$^{-/-}$* embryos at the day 10 post serum starvation were harvested by centrifugation at 2000 rpm for 5 min at 4 °C. The cells were resuspended in 2.4 ml of deciliation buffer (20 mM PIPES, 250 mM sucrose, 20 mM CaCl$_2$, 0.05% Triton X-100, pH 5.5) and agitated vigorously for 10 min on a vortex mixer (G560OE, Scientific Industries). After centrifugation at 600 × g for 5 min at 4 °C, the cilia-containing supernatant was collected by centrifugation at 20,000 × g for 30 min. The pelleted cilia were washed

twice with 1 ml of PBS, followed by centrifugation at 20,000 × g for 15 min. The isolated cilia were lysed in 100 µl of lysis buffer [20 mM Tris-HCl (pH 7.5), 100 mM KCl, 0.1% NP-40, 1 mM EDTA, 10 mM Na$_4$O$_7$P$_2$, and protease inhibitors] and boiled at 100 °C for 10 min. 50 µl of the samples was used for Mass spectrometry analysis and another 50 µl was used for immunoblotting (Fig. 2d). The full scan blots are provided in the Source Data file.

**Label-free quantitative mass spectrometry**. In all, 50 µL of lysed wild type and *Wdr47$^{-/-}$* cilia samples were precipitated with acetone. The protein pellet was dried by using a Speedvac for 1–2 min. The pellet was subsequently dissolved in 8 M urea, 100 mM Tris-HCl, pH 8.5. TCEP (final concentration is 5 mM) (Thermo Scientific) and iodoacetamide (final concentration is 10 mM) (Sigma) for reduction and alkylation were added to the solution and incubated at room temperature for 30 min, respectively. The protein mixture was diluted four times and digested overnight with Trypsin at 1:50 (w/w) (Promega). The tryptic-digested peptide solution was desalted using a MonoSpin$^{TM}$ C18 column (GL Science, Tokyo, Japan) and dried with a SpeedVac.

The peptide mixture was analyzed by a home-made 30-cm-long pulled-tip analytical column (75 µm ID packed with ReproSil-Pur C18-AQ 1.9 µm resin, Dr. Maisch GmbH), the column was then placed in-line with an Easy-nLC 1200 nano HPLC (Thermo Scientific, San Jose, CA) for mass spectrometry analysis. The analytical column temperature was set at 55 °C during the experiments. The mobile phase and elution gradient used for peptide separation were as follows: 0.1% formic acid in water as buffer A and 0.1% formic acid in 80% acetonitrile as buffer B, 0–1 min, 3–8% buffer B; 1–301 min, 8–25% buffer B; 301–339 min, 25–50% buffer B, 339–340 min, 50–100% buffer B, 340–360 min, 100% buffer B. The flow rate was set as 300 nl/min.

Data-dependent tandem mass spectrometry (MS/MS) analysis was performed with a Q Exactive Orbitrap mass spectrometer (Thermo Scientific, San Jose, CA). A cycle of one full-scan MS spectrum (m/z 300–1800) was acquired followed by top 20 MS/MS events, sequentially generated on the first to the twentieth most intense ions selected from the full MS spectrum at a 28% normalized collision energy. Full scan resolution was set to 70,000 with automated gain control (AGC) target of 3e$^6$. MS/MS scan resolution was set to 17,500 with isolation window of 1.8 *m/z* and AGC target of 1e$^5$. The number of microscans was one for both MS and MS/MS scans and the maximum ion injection time was 50 and 100 ms, respectively. The dynamic exclusion settings used were as follows: charge exclusion, 1 and >8; exclude isotopes, on; and exclusion duration, 15 s. MS scan functions and LC solvent gradients were controlled by the Xcalibur data system (Thermo Scientific).

MS/MS data were processed using Maxquant software version V1.6.10.43. MS/MS spectra were searched by the Andromeda search engine against the SwissProt Mouse database at a false discovery cutoff ≤1%. Data were searched at 20 ppm mass tolerances for precursor ions for mass calibration and six amino acids were required as the minimum peptide. LFQ intensity was used as relative quantification of protein.

**Light microscopy**. mEPCs grown on fibronectin-coated 29-mm glass-bottom dishes (Cellvis, D29-14-1.5-N) were fixed with 4% paraformaldehyde in PBS for 10 min at room temperature and permeabilization with 0.5% Triton X-100 for 15 min. In Figs. 1e and 4c, mEPCs were pre-extracted with 0.5% Triton X-100 for 30 s before fixation. GFP signals in mEPCs were visualized by immunostaining using anti-GFP antibody. All the antibodies used are listed in Supplementary Table 2.

Confocal images were captured by using Leica TCS SP8 system with a ×63/1.40 oil immersion objective and Z-stack images were obtained with maximum intensity projections. 3D-SIM images were captured with Delta Vision OMX SR imaging system (GE Healthcare) equipped with a Plan Apo ×60/1.42 NA oil-immersion objective lens (Olympus). Serial Z-stack sectioning was performed at 125-nm intervals. Images were processed with SoftWoRx software.

Ciliary motilities were recorded at 140 fps (frames per second) by using an Andor Neo sCMOS camera on Olympus IX71 microscope with a ×63/1.40 oil immersion objective[43]. To track fluid flows driven by ciliary beat[82], fluorescent beads (Fluoresbrite PolyFluor 570 microspheres, Polysciences, 24061-10) were added at 1:200 dilution to the culture medium and imaged at 20 fps for 30 s with a ×63/1.40 oil immersion objective on an Olympus IX81 equipped with Hamamatsu EMCCD camera. Images were processed with ImageJ and Adobe Photoshop CS6.

**Electron microscopy**. For transmission EM, mouse trachea or cultured mEPCs were fixed in 2.5% glutaraldehyde overnight at 4 °C, washed with PBS, and treated with 1% OsO4 for 30 min at room temperature. The samples were dehydrated with graded ethanol series and embedded in Epon 812 resin. In total, 70-nm ultrathin sections were stained with 1% lead citrate and 2% uranyl acetate. Images were captured at 80 KV using a Tecnai G2 Spirit transmission electron microscope (FEI, Hillsboro, OR).

For scanning EM, the cortex of P10 mouse brains was fixed in 2.5% glutaraldehyde and 2% paraformaldehyde overnight at 4 °C and dehydrated with a graded ethanol series. Critical point drying was carried out before metal shadowing. The images were taken with a FEI Quanta 250 scanning electron microscope.

**Magnetic resonance imaging, tissue section and H&E staining**. Mice were anesthetized with isoflurane. T2 weighted spin echo images of the head were acquired by using BioSpec 70/30 USR (Bruker). The imaging parameters were: slice thickness 0.5 mm; time of repetition (TR) 2777.2 ms; and time of echo (TE) 34 ms.

P10 mouse brains were dissected and sectioned into 250-μm-thick sagittal slices using a Leica VT 1000S vibratome. Images were captured with an Olympus SZX16 Stereo Microscope.

For H&E staining, P14 mouse brains and noses were paraffin-embedded and sectioned into 5-μm thick slices with an RM 2235 microtome (Leica). The sections were deparaffinized with xylene, rehydrated, and stained with hematoxylin for 5 min and eosin for 1 min. The tissue images were captured with an Olympus BX51 microscope.

**Immunoprecipitation**. Immunoprecipitation experiments were performed as described[78]. Cells were lysed with lysis buffer [20 mM Tris-HCl, pH 7.5, 100 mM KCl, 0.1% NP-40, 1 mM EDTA, 10 mM Na$_4$O$_7$P$_2$, 10% Glycerol, and protease inhibitors (Sigma, 539134)] and was cleared by centrifugation at $14,000 \times g$ for 10 min at 4 °C. The precleared cell lysates were incubated with 20 μl of anti-FLAG beads (Sigma, A2220) for 2 h. The beads were washed three times with lysis buffer and three times with wash buffer (20 mM Tris-HCl, pH 7.5, 150 mM KCl, 0.5% NP-40, 1 mM EDTA, 10 mM Na$_4$P$_2$O$_7$,10% Glycerol). The proteins on the FLAG beads were eluted with 30 μl of 1 mg/ml FLAG peptide.

**Quantification and statistics**. Microscopic and biochemical results were repeated at least twice. Quantification results are presented as mean ± s.d. unless otherwise stated. Two-sided Student's $t$ test (GraphPad Prism software) was used to calculate P-values between unpaired samples. Differences were considered significant when $P < 0.05$. Only results from three or more independent experiments were applied to the $t$-tests.

**Reporting summary**. Further information on research design is available in the Nature Research Reporting Summary linked to this article.

## Data availability

Raw data of label-free quantitative (LFQ) proteomic analysis results (Fig. 2c) have been deposited to the ProteomeXchange Consortium (http://proteomecentral.proteomexchange.org) via the iProX partner repository[83] with the accession code (PXD028219). Source data are provided with this paper. Any remaining data that support the results of this study are available from the corresponding author upon reasonable request. Source data are provided with this paper.

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

## Acknowledgements

The authors thank institutional core facilities for cell biology and molecular biology for technical supports. We also thank Dr. Kai Jiang (Wuhan University) and Dr. Wenxiang Meng (Institute of Genetics and Developmental Biology, Chinese Academy of Sciences) for plasmids and antibodies. This work was supported by National Key R&D Program of China (2017YFA0503500), National Natural Science Foundation of China (31991192, 31771495, and 31900503), and Chinese Academy of Sciences (XDB19000000).

## Author contributions

X.Z. and X.Y. conceived and directed the project; H.L., J.-Q.Z. and L.Z. performed major experiments; Y.C., Y.Z., L.X. and W.Z. contributed results; Y.Y. and C.P. performed the label-free quantitative (LFQ) proteomic analysis; J.Z. provided 3D-SIM; X.Z., X.Y., H.L., J.-Q.Z. and L.Z. designed experiments, interpreted data, and wrote the paper.

## Competing interests

The authors declare no competing interests.
