## [Peer Review File · Nature Communications]

Reviewers' Comments:

Reviewer #1:

Remarks to the Author:

Most of motile cilia have 9+2 configuration, which is composed of nine pairs of peripheral MTs and a central pair (CP) of MTs. CP is not required for motility of cilia per se but regulates their motion pattern. How CP is formed is poorly understood, despite that several genes are known to be involved in CP formation. In this paper, the authors employ mouse genetics combined with ependymal cell culture, and report convincing data for the role of three CP-localized proteins, Wdr47, Camsap and Katanin, in CP formation. I believe that the paper provides important information for our understanding of how ciliary motility is regulated and how non-centrosomal MTs are arrayed. I only have minor comments.

1) It is clear that CP MTs are lost in ependymal cilia and trachea cilia from Wdr47(-/-) mouse. However, an electron-dense structure remain at the center of mutant ependymal cilia (Fig. 1i), while such a structure is not seen in mutant trachea cilia (Fig. S2). The author need to comment on this structure: how often is it detected (not seen in Fig. 2i), dependent on the proximal-distal level, undetected in ependymal cilia with GFP-p60(K257A) (as in Fig. 7e)?

2) While most of motile cilia have 9+2 configuration, node cilia that generate the directional fluid flow for left-right asymmetry are 9+0 lacking the central pair. I wonder if Wdr47 (-/-) mouse exhibits laterality defects.

Reviewer #2:

Remarks to the Author:

In this manuscript, Liu et al. reveals that Wdr47, a WD40 repeat-containing protein associated indirectly with microtubules (MTs), is essential for the growth and maintenance of the central pair (CP) MT of motile cilia. This work identifies novel players for the CP MT formation, including Wdr47, Camsaps, and Katanin. They show that Wdr47 and Camsaps localize at the ciliary lumen. The part of this work related to Wdr47 is complete and solid. They show that Wdr47 is concentrated at the ciliary tip of short cilia, suggesting its role in the early stage of CP formation. Depletion of Wdr47 abolishes CP formation, destroys the planar beating pattern of cilia, changes CP protein localization, and affects CP structure and function even after the CP is formed. Wdr47-deficient mice display PCD-like phenotypes. For Camsaps including Camsap1, Camsap2, and Camsap3, previously it was shown that Wdr47 can be recruited to MT minus ends by Camsaps. They find that Camsaps localization in cilia is abolished when depleting Wdr47. Overexpression of Camsap 1, 2, and 3 rescue CP formation and beating pattern in Wdr47-/- cells, although Camsap2 rescues only 5-10% to the planar beating pattern and ~10% of CP formation. Wdr47 overexpression in Wdr47-/- rescues the localization of Camsap1. They study the contribution of different domains of Wdr47 (WdrN and WdrC) and different domains of Camsaps (N1, N2, CKK, LC). They find that the CKK domain of Camsaps is enough for planar beating without any need of Wdr47. This conclusion clearly shows that Camsaps, at least Camsap1, require Wdr47 for its essential functions on CP formation and planar beating. For Katanin, they find that p60 catalytic subunit of MT severing protein Katanin localizes to a region of cilia. Dominant inhibiting p60_K257A impairs planar beating and CP formation, even in the Camsap1 rescue Wdr47-/- cells. That is, Katanin is also an important element for CP formation and function.

Major comments:

- There are obvious differences among Camsap1, 2, and 3 in several aspects through the article. For example, the patterns of Camsap1, 2, and 3 are not similar in Fig 4b. The authors should clearly state their differences and avoid lumping all results into Camsaps.
- The authors show that Camsap3 is most abundant in mass spec results of ciliary proteins. It is unclear why Camsap3 is not focused in the study but instead the low abundant Camsap1, which does not interact with Katanin, is focused in this study. The authors may want to elaborate more on this.
- Images of Wdr47 in three different experiments of Figure 4c are quite different. Thus, the differences of Camsap1, 2, and 3 seem to be minimized. The authors should find images with

similar exposure for all three cases of Wdr47 so that at least their images look similar and know whether the differences of Camsap1, 2, and 3 should be addressed.

- Line 246-247: This conclusion does not reflect what is observed. Camsap1 localization is not well rescued by WdrC or WdrN, whereas the super-resolution result shows the lumen occupancy of WdrC, not Camsap1. It is misleading to state this sentence here. And thus, the subtitle in line 234 should be modified accordingly.

- Although it is clear that Katanin is important for CP formation, more experiments need to be conducted to explicitly state that Katanin cooperates with Wdr47 and Camsaps for CP formation, as those stated in the title, the abstract, and the discussion, especially that Camsap1 does not interact with Katanin and the experiments done with Katanin in this manuscript are all in Camsap1.

- For the Katanin experiment, GFP-p60-WT should be included.

Minor comments:

- Line 152: the meaning of the sentence "suggesting depletion of Wdr47 prior to CP formation" is unclear.

- Line 235: Fig 5a-g is about the effect of Camsaps on CP, not directly Wdr47. The authors should modify this sentence.

- Line 242: Should change Wdr47N to WdrN to be consistent.

- The result of the WdrC and WdrN part does not seem to bring to clear conclusion. For example, neither WdrC nor WdrN is able to rescue CP formation, planar beating, or Camsap1 localization. The authors should explain the result more clearly.

- Lines 261, 266, 269: should change mutants to constructs

- Line 310 and other places: sever, not server

- Comparing Fig 5c and Fig 6e, it seems that Ca2LC and Ca2CKK rescue the planar beating much better than Camsap2. Please explain.

- The figure labeling should be made clearer. Specifically, the inserts and sub-panels usually are not clearly labeled or colored, such as those in Figs. 7c, 4e, 4c.

- It appears that the CKK domain of Camsap1 is enough for CP formation and planar beating. The authors may want to discuss the recruitment of Wdr47 by Camsaps to MT or the recruitment of Camsaps by Wdr47 to MT for the comparison of the results of this work and previous papers.

Response to reviewers' comments

Reviewer #1:

Most of motile cilia have 9+2 configuration, which is composed of nine pairs of peripheral MTs and a central pair (CP) of MTs. CP is not required for motility of cilia per se but regulates their motion pattern. How CP is formed is poorly understood, despite that several genes are known to be involved in CP formation. In this paper, the authors employ mouse genetics combined with ependymal cell culture, and report convincing data for the role of three CP-localized proteins, Wdr47, Camsap and Katanin, in CP formation. I believe that the paper provides important information for our understanding of how ciliary motility is regulated and how non-centrosomal MTs are arrayed. I only have minor comments.

Response:

We thank our reviewer for appreciating our efforts and helping us to improve the manuscript.

1) It is clear that CP MTs are lost in ependymal cilia and trachea cilia from *Wdr47*(-/-) mouse. However, an electron-dense structure remain at the center of mutant ependymal cilia (Fig. 1i), while such a structure is not seen in mutant trachea cilia (Fig. S2). The author need to comment on this structure: how often is it detected (not seen in Fig. 2i), dependent on the proximal-distal level, undetected in ependymal cilia with GFP-p60(K257A) (as in Fig. 7e)?

Response:

We appreciate the comments of our reviewer. We re-examined the TEM images of *Wdr47*^{-/-} cilia and found that the electron dense materials tended to emerge at the center of distal sections (71%) as compared to proximal sections (23%), when surrounding microvilli were used to distinguish the proximal ciliary region from the distal region. Similar observations have been reported for CP-less flagella of some *Chlamydomonas* mutants (Adams et al., 1981; Witman et al., 1978). Such phenomena are possibly due to remnants of CP components in the absence of CP MTs, though details require future investigation. We have included this information in the Results section (page 6, lines 127-132) of the revised manuscript and modified the text accordingly.

2) While most of motile cilia have 9+2 configuration, node cilia that generate the directional fluid flow for left-right asymmetry are 9+0 lacking the central pair. I wonder if *Wdr47* (-/-) mouse exhibits laterality defects.

Response:

Wdr47^{-/-} mice die of suffocation shortly after birth (Chen et al., 2020) but we have not

observed any laterality defects. Nor are such defects reported by other groups (Kannan et al., 2017). Therefore, Wdr47 probably only affects cilia with a 9+2 axoneme.

Reviewer #2:

In this manuscript, Liu et al. reveals that Wdr47, a WD40 repeat-containing protein associated indirectly with microtubules (MTs), is essential for the growth and maintenance of the central pair (CP) MT of motile cilia. This work identifies novel players for the CP MT formation, including Wdr47, Camsaps, and Katanin. They show that Wdr47 and Camsaps localize at the ciliary lumen. The part of this work related to Wdr47 is complete and solid. They show that Wdr47 is concentrated at the ciliary tip of short cilia, suggesting its role in the early stage of CP formation. Depletion of Wdr47 abolishes CP formation, destroys the planar beating pattern of cilia, changes CP protein localization, and affects CP structure and function even after the CP is formed. Wdr47-deficient mice display PCD-like phenotypes. For Camsaps including Camsap1, Camsap2, and Camsap3, previously it was shown that Wdr47 can be recruited to MT minus ends by Camsaps. They find that Camsaps localization in cilia is abolished when depleting Wdr47. Overexpression of Camsap 1, 2, and 3 rescue CP formation and beating pattern in Wdr47^{-/-} cells, although Camsap2 rescues only 5-10% to the planar beating pattern and ~10% of CP formation. Wdr47 overexpression in Wdr47^{-/-} rescues the localization of Camsap1. They study the contribution of different domains of Wdr47 (WdrN and WdrC) and different domains of Camsaps (N1, N2, CKK, LC). They find that the CKK domain of Camsaps is enough for planar beating without any need of Wdr47. This conclusion clearly shows that Camsaps, at least Camsap1, require Wdr47 for its essential functions on CP formation and planar beating. For Katanin, they find that p60 catalytic subunit of MT severing protein Katanin localizes to a region of cilia. Dominant inhibiting p60_K257A impairs planar beating and CP formation, even in the Camsap1 rescue Wdr47^{-/-} cells. That is, Katanin is also an important element for CP formation and function.

Major comments:

1) There are obvious differences among Camsap1, 2, and 3 in several aspects through the article. For example, the patterns of Camsap1, 2, and 3 are not similar in Fig 4b. The authors should clearly state their differences and avoid lumping all results into Camsaps.

Response:

Thanks for the comments. In the revised manuscript, we have carefully modified the main text concerning similarities and differences among Camsap1, 2, and 3 and used

Camsaps only when discriminations are not required.

Furthermore, for a more thorough comparison we also examined the rescue effects of the LC and CKK constructs of Camsap2 and -3 on Hydin and central MTs and have combined the results into Fig. 6f-h and Supplementary Fig. 7 in the revised manuscript. Similar to the LC and CKK constructs of Camsap1, the corresponding constructs of Camsap2 and -3 localized to multicilia in *Wdr47*-deficient mEPCs and were able to prominently restore the ciliary localization of Hydin (Fig. 6f and Supplementary Fig. 7). They also substantially rescued the central MT formation, though detailed efficiencies varied (Fig. 6g, h). Our results altogether suggest that Camsap1-3 share similar functions in the central MT formation, though their detailed contributions may vary. We have also modified the Discussion section and discussed their functional similarity and difference in more details.

2)The authors show that Camsap3 is most abundant in mass spec results of ciliary proteins. It is unclear why Camsap3 is not focused in the study but instead the low abundant Camsap1, which does not interact with Katanin, is focused in this study. The authors may want to elaborate more on this.

Response:

We appreciate the suggestion of our reviewer. We actually do not intend to only focus on Camsap1. We have presented results on subcellular localizations (Fig. 4b, c) and rescue effects of all Camsaps (Fig. 5a-g). As described in our response to the comment #1, we have included rescue results of the LC and CKK constructs of Camsap2 and -3 in the revised manuscript (Fig. 6f-h and Supplementary Fig. 7). To improve the clarity, we have further clarified why we extensively examined the GFP-Camsap1 samples (i.e., three biological repeats, n=805 cross-sections) by EM (page 10, lines 228-230). We have also presented the EM results of the GFP-Camsap3 sample in a pie chart (Fig. 5g) in addition to the description in the main text for a better clarity.

During the revision, we noticed that a recent paper (Robinson et al., 2020) reports that Camsap3 localizes to the base of axonemes in murine nasal multiciliated cells and *Camsap3* deficiency results in the CP loss in 77.4% axonemal cross-sections of the cells. Consistently, *Camsap3*-deficient mice display severe nasal airway blockage, similar to our *Wdr47*^{Kof/-} mice (Fig. 3b). In sharp contrast to the severe hydrocephalus of our *Wdr47*^{lox/lox}; *GFAP-Cre* cKO mice (Fig. 3c-e), *Camsap3*-deficient mice only display mild or no hydrocephalus phenotypes (Robinson et al., 2020), implying a relatively weak influence on ependymal multicilia. On the other hand, multicilia-related defects have not been documented for *Camsap1*- or *Camsap2*-deficient mice (Pongrakhananon et al., 2018; Zhou et al., 2020). These publications demonstrate the importance of Camsap3 in the CP formation of mainly nasal multicilia and echo our proposed redundant roles of Camsaps. It thus remains to be shown whether the depletion of all Camsaps leads to the complete CP loss as in our case of *Wdr47* deficiency

(Figs 1, 2, and Supplementary Figs 2, 3). As the expression levels of individual Camsaps may vary in multiciliated cells of different tissues, combinations of double knockout mice may be used to assess their tissue-specific contributions as well. We have included similar discussions in the revised manuscript.

3) Images of Wdr47 in three different experiments of Figure 4c are quite different. Thus, the differences of Camsap1, 2, and 3 seem to be minimized. The authors should find images with similar exposure for all three cases of Wdr47 so that at least their images look similar and know whether the differences of Camsap1, 2, and 3 should be addressed.

Response:

As the expression levels of GFP-Camsap1 generally exceeded those of GFP-Camsap2 or -Camsap3 (Fig. 5a), fluorescent signals of both Wdr47 and GFP at the ciliary tip tended to be more prominent in GFP-Camsap1-expressing mEPCs than in cells expressing GFP-tagged Camsap2 or -3. Following the request of our reviewer, we have replaced the cells in Fig. 4c with cells of similar total GFP intensity, imaged with identical exposure time and laser intensity, in the revised manuscript. Similar to the previous images, all three GFP-tagged Camsaps still tended to be enriched at the tip of short cilia with Wdr47 (Fig. 4c, revised manuscript).

4) Line 246-247: This conclusion does not reflect what is observed. Camsap1 localization is not well rescued by WdrC or WdrN, whereas the super-resolution result shows the lumen occupancy of WdrC, not Camsap1. It is misleading to state this sentence here. And thus, the subtitle in line 234 should be modified accordingly.

Response:

We are sorry for causing the misunderstanding. As the partial rescue effects of Camsaps (Fig. 5a-g) suggested an essential role of Wdr47 for efficient production and proper positioning of central MTs, Fig. 5h-k are presented to clarify whether Wdr47 functions by increasing the regional ciliary concentration of Camsaps. The rescue of ciliary Camsap1 localization by GFP-tagged Wdr47 but not WdrC or WdrN (Fig. 5i) indicates that Wdr47 requires both Camsap-interacting and cilia-localization regions for the ciliary enrichment of endogenous Camsaps (please also refer to Fig. 4d-f, in which we showed that *Wdr47* deficiency impaired the ciliary localizations of endogenous Camsap1-3). Furthermore, as *Wdr47*^{-/-} multicilia are CP-less (Fig. 1i), the super-resolution result (Fig. 5k) suggests that Wdr47 uses its C-terminal region (WdrC) to target to the ciliary central lumen independently of central MTs. Taken together, we conclude that Wdr47 binds to Camsaps through its N terminal region and targets them to the ciliary central lumen through its C-terminal region for

efficient and proper central MT formation.

In the revised manuscript, we have modified this part of text to improve the clarity. We have also modified the subtitle as “Wdr47 recruits Camsaps to the central lumen for proper CP formation”.

5) Although it is clear that Katanin is important for CP formation, more experiments need to be conducted to explicitly state that Katanin cooperates with Wdr47 and Camsaps for CP formation, as those stated in the title, the abstract, and the discussion, especially that Camsap1 does not interact with Katanin and the experiments done with Katanin in this manuscript are all in Camsap1.

Response:

We appreciate the comments. As requested, we performed additional experiments. We found that endogenous Katanin p60 also displayed ciliary localization in mEPCs and the tip enrichment in short cilia (Fig. 7a). As described below (please refer to our response to comment #6), we also examined the effects of p60 overexpression (Fig. 7c-g). Overexpression of GFP-p60 mildly increased the percentage of wild-type mEPCs with aberrant ciliary beat as compared to Centrin1-GFP (Fig. 7e); EM analyses revealed increased incidence of axonemal cross-sections lacking the CP or containing aberrant numbers of central MTs (Fig. 7f). Therefore, appropriate levels of Katanin are also important for proper central MT formation. These results further strengthen our model (Fig. 7h) suggesting that Katanin, Camsaps, and Wdr47 function together for proper central MT formation: Katanin is needed to generate MT seeds, Camsaps bind and stabilize the MT minus ends, and Wdr47 targets Camsaps to the ciliary central lumen.

As explained in our response to comment #2, we have included additional results on Camsap2 and -3 in the revised manuscript for a more thorough comparison among the three Camsaps. We coexpressed GFP-Camsap1 and RFP-p60/p60^{K257A} in *Wdr47*^{-/-} mEPCs (Fig. 7g, revised manuscript) for two reasons. Firstly, in our hand GFP-Camsap1 displayed the highest expression levels and best rescue effects in *Wdr47*^{-/-} mEPCs (Fig. 5a-g). Therefore, it is obviously the best choice for further assessment of Katanin function. Secondly, because Camsap1 does not interact with Katanin, it allowed us to clearly attribute the effects of p60/p60^{K257A} to Katanin. The use of Camsap3 would complicate the situation. For instance, we would need to confirm first that p60 and p60^{K257A} would not display different binding affinity to Camsap3 and differentially affect its MT-binding and stabilizing activities.

In addition, the two paralogues of p60, Katnal1 and Katnal2, which have also been shown to affect the growth and motility of multicilia (Banks et al., 2018; Willsey et al., 2018), might also have a role in the central MT seeds production because the overexpression of p60^{K257A} might inhibit Katnal1 and Katnal2 as well. Although Camsap1 does not interact with p60/p80, whether it could bind to Katnal1 or Katnal2 is unknown. Future investigations are

thus still required to clarify the detailed interplays among Wdr47, Camsaps, and Katanin. We have modified the discussion section accordingly.

6) For the Katanin experiment, GFP-p60-WT should be included.

Response:

Following the request, we re-performed all the assays to include wildtype GFP-p60 and have presented the new results in Fig. 7c-g in the revised manuscript. Live imaging indicated that the overexpression of GFP-p60 mildly increased the percentage of wild-type mEPCs with aberrant ciliary beat as compared to Centrin1-GFP, but the effect was less striking than that of GFP-p60^{K257A} (Fig. 7e). EM analyses revealed increased incidence of CP-less cross-sections (14%) in the GFP-p60 samples comparing to the value (6%) of the Centrin1-GFP samples (Fig. 7f). Notably, cross-sections containing one (3%) and >2 (4%) central MTs emerged in the GFP-p60 samples, which were not observed in either the Centrin1-GFP or the GFP-p60^{K257A} samples (Fig. 7f). In comparison, co-expression of RFP-p60 did not significantly alter the rescue effects of GFP-Camsap1 as RFP-p60^{K257A} did in *Wdr47*-deficient mEPCs (Fig. 7g). These results further corroborate our proposed role of Katanin in central MT formation.

Minor comments:

1) Line 152: the meaning of the sentence “suggesting depletion of Wdr47 prior to CP formation” is unclear.

Response:

We have modified the sentence to “suggesting that under the condition Wdr47 was depleted prior to the CP formation in the cells” in the revised manuscript (page 7, line 159-160). We have also modified the paragraph to improve the clarity of presentation.

2) Line 235: Fig 5a-g is about the effect of Camsaps on CP, not directly Wdr47. The authors should modify this sentence.

Response:

We have modified the sentence to “as the partial rescue effects of overexpressed Camsaps (Fig. 5a-g) suggested a role of Wdr47 for efficient production and proper positioning of central MTs, we speculated...” (page 10, line 246-247).

3) Line 242: Should change Wdr47N to WdrN to be consistent.

Response:

We have corrected the typo in the revised manuscript.

4) The result of the WdrC and WdrN part does not seem to bring to clear conclusion. For example, neither WdrC nor WdrN is able to rescue CP formation, planar beating, or Camsap1 localization. The authors should explain the result more clearly.

Response:

As explained in our response to major comment #4, we have modified the text to improve the clarity of presentation in the revised manuscript.

5) Lines 261, 266, 269: should change mutants to constructs

Response:

The requested changes have been made in the revised manuscript.

6) Line 310 and other places: sever, not server

Response:

We are sorry for the mistakes. We have corrected them in the revised manuscript.

7) Comparing Fig 5c and Fig 6e, it seems that Ca2LC and Ca2CKK rescue the planar beating much better than Camsap2. Please explain.

Response:

As described in our response to comment #1, for a more thorough comparison we also examined the rescue effects of the LC and CKK constructs of Camsap2 and -3 on Hydin and central MTs (Fig. 6f-h and Supplementary Fig. 7 in the revised manuscript). The different effects of Camsaps could be due to their individual differences in regulating MT dynamics and possibly to the differences in exogenous expression levels as well. Future studies are required to clarify these issues. In response to the request of our reviewer, we have included a paragraph to discuss the similarity and difference among the three Camsaps (please refer to page 15, 2nd paragraph).

8) The figure labeling should be made clearer. Specifically, the inserts and sub-panels usually are not clearly labeled or colored, such as those in Figs. 7c, 4e, 4c.

Response:

In the revised manuscript, we have re-labelled the insets to improve the clarity.

9) It appears that the CKK domain of Camsap1 is enough for CP formation and planar beating. The authors may want to discuss the recruitment of Wdr47 by Camsaps to MT or the recruitment of Camsaps by Wdr47 to MT for the comparison of the results of this

work and previous papers.

Response:

We appreciate the comments of our reviewer. Previous publications including ours report that Wdr47 alone does not bind to MTs (Chen et al., 2020; Wang et al., 2012). Nevertheless, it can be recruited to cellular MTs by Camsap3 (Chen et al., 2020). We have described this in the manuscript (page 8, line 192-193, revised manuscript).

References:

- Adams, G.M., Huang, B., Piperno, G., and Luck, D.J. (1981). Central-pair microtubular complex of *Chlamydomonas* flagella: polypeptide composition as revealed by analysis of mutants. *J Cell Biol* *91*, 69-76.
- Banks, G., Lassi, G., Hoerder-Suabedissen, A., Tinarelli, F., Simon, M.M., Wilcox, A., Lau, P., Lawson, T.N., Johnson, S., Rutman, A., *et al.* (2018). A missense mutation in *Katnal1* underlies behavioural, neurological and ciliary anomalies. *Mol Psychiatr* *23*, 713-722.
- Chen, Y., Zheng, J., Li, X., Zhu, L., Shao, Z., Yan, X., and Zhu, X. (2020). Wdr47 Controls Neuronal Polarization through the Camsap Family Microtubule Minus-End-Binding Proteins. *Cell Rep* *31*, 107526.
- Kannan, M., Bayam, E., Wagner, C., Rinaldi, B., Kretz, P.F., Tilly, P., Roos, M., McGillewie, L., Bar, S., Minocha, S., *et al.* (2017). WD40-repeat 47, a microtubule-associated protein, is essential for brain development and autophagy. *Proc Natl Acad Sci U S A* *114*, E9308-E9317.
- Pongrakhananon, V., Saito, H., Hiver, S., Abe, T., Shioi, G., Meng, W., and Takeichi, M. (2018). CAMSAP3 maintains neuronal polarity through regulation of microtubule stability. *Proc Natl Acad Sci U S A* *115*, 9750-9755.
- Robinson, A.M., Takahashi, S., Brotslaw, E.J., Ahmad, A., Ferrer, E., Procissi, D., Richter, C.P., Cheatham, M.A., Mitchell, B.J., and Zheng, J. (2020). CAMSAP3 facilitates basal body polarity and the formation of the central pair of microtubules in motile cilia. *Proc Natl Acad Sci U S A* *117*, 13571-13579.
- Wang, W., Lundin, V.F., Millan, I., Zeng, A., Chen, X., Yang, J., Allen, E., Chen, N., Bach, G., Hsu, A., *et al.* (2012). Nemitin, a novel Map8/Map1s interacting protein with Wd40 repeats. *PLoS One* *7*, e33094.
- Willsey, H.R., Walentek, P., Exner, C.R.T., Xu, Y.X., Lane, A.B., Harland, R.M., Heald, R., and Santama, N. (2018). Katanin-like protein *Katnal2* is required for ciliogenesis and brain development in *Xenopus* embryos. *Dev Biol* *442*, 276-287.
- Witman, G.B., Plummer, J., and Sander, G. (1978). *Chlamydomonas* flagellar mutants lacking radial spokes and central tubules. Structure, composition, and function of specific axonemal components. *J Cell Biol* *76*, 729-747.
- Zhou, Z., Xu, H., Li, Y., Yang, M., Zhang, R., Shiraishi, A., Kiyonari, H., Liang, X., Huang, X., Wang, Y., *et al.* (2020). CAMSAP1 breaks the homeostatic microtubule network to instruct neuronal polarity. *Proc Natl Acad Sci U S A* *117*, 22193-22203.

Reviewers' Comments:

Reviewer #1:

Remarks to the Author:

The authors have adequately addressed my previous comments. As far as I am concerned, the paper can be accepted.

Reviewer #3:

Remarks to the Author:

The manuscript from Liu et al. sheds some light on the mechanism of formation of the central pair of microtubules of motile cilia in ependymal cells in the mouse. Overall, this is an interesting manuscript characterized by a substantial degree of technical complexity.

I agree with Reviewer #2 that "the part of this work related to Wdr47 is complete and solid", but I also agree that the portion of the manuscript related to the role of CAMSAPs and the role of Katanin presents some weaknesses. These are not fully addressed in the rebuttal precluding the manuscript from publication in Nature Communications in its current form.

I believe that the manuscript might be interesting enough for publication in Nature Communication in a heavily revised form, but the authors should modify the manuscript substantially both in the results and discussion sections to clarify the potentially different mechanistic roles of Camsap1 and 3 in CP formation and their role in cooperating with Katanin and keep the focus of their paper on Wdr47 data. I am sorry that my review cannot be more positive, as it is clear that the work required a major effort from the authors and already went through a round of review.

This reviewer is not convinced by the data suggesting that all Camsaps have a direct role in CP formation, they are CP proteins and co-localize with Wdr47 and found that area of the manuscript confusing.

- Camsaps show very different localization patterns in Fig.4c, some not convincingly consistent with a role in central pair formation. For example, Camsap3 is localized at the base of motile cilia as shown in Fig. 4b, away from Hydin a known central pair protein. How does Camsap3 play a role in CP formation when its subcellular localization is away from the CP region? Results presented previously in Robinson et al. PNAS June 2020 show that Camsap3 has a dual localization at basal bodies and at the base of cilia (possibly the TZ), which is consistent with a role in CP formation. The localization data presented here seem different and not consistent with a role in CP formation.

In addition, Camsap1 also only shows minimal localization to the base of the CP in short cilia and is largely concentrated at the tip of growing motile cilia. How does this subpopulation of Camsap1 contribute to CP formation if at all? Assuming that only the small portion associated at the base of cilia of (some) of the Camsaps plays a role in CP formation, what kinds of role do the different Camsaps play in CP formation?

- Data presented in Fig. 4c show Camsaps have different localization but somehow they always colocalize with Wdr47. The data presented in Figs. 4c and 7c are not convincing, the authors should swap the secondary antibodies colors of Ac-Tub (in blue) and the red channel (WDR47 for 4c and Camsap1 for 7c) to make sure there is no bleed through from GFP.

- Overexpression of GFP-Camsap3 rescues only ~20% of the motile cilia for recruitment of Hydin in the WDR47 -/- background, but rescue 75% of CP defects by TEM. Can the author look at other component of the CP for rescue in the GFP-Camsap3 overexpressing cells to check if this difference in the results is due to using different assays or due to a specific effect on HYDIN?

- The authors show that Camsap1 and 3 overexpression convincingly rescue the phenotype on CP formation of Wdr47 -/- cells. As pointed out by reviewer 2, GFP-Camsap2 rescue very mildly Wdr47 -/- CP formation phenotype, so might have just an indirect role in CP formation driven by Wdr47.

As the authors suggest, there might be a certain degree of redundancy in the function of Camsaps such that overexpression might rescue some of the phenotype of Wdr47 caused by one of the Camsaps that is involved in CP formation. If this is correct, this work does not yet clarify which Camsap is relevant in CP formation. Alternatively, they might all play different roles in CP formation, but these different roles are not addressed.

- The data on Katanin are interesting, but they don't justify the claims from the authors of an interplay between Wdr47, Camsaps and Katanin in CP formation. As discussed by Reviewer 2 "more experiments need to be conducted to explicitly state that Katanin cooperates with Wdr47 and Camsaps for CP formation, as those stated in the title, the abstract, and the discussion..." The interplay statement should be toned down substantially in the manuscript and the discussion should clearly be focused on Camsap1 as the authors show only data for this protein. In the discussion the authors need to better elaborate on what is known about Camsap3 and Katanin interaction from published work on Tetrahymena. Ideally, they should compare in their system the roles of Camsap 1 and 3 in the interplay with Katanin function, even if the experiment would have some limitation as discussed in the rebuttal.

Response to Reviewer comments

Reviewer #1:

The authors have adequately addressed my previous comments. As far as I am concerned, the paper can be accepted.

Response:

We thank our reviewer for appreciating our work and helping us to improve our manuscript.

Reviewer #3:

The manuscript from Liu et al. sheds some light on the mechanism of formation of the central pair of microtubules of motile cilia in ependymal cells in the mouse. Overall, this is an interesting manuscript characterized by a substantial degree of technical complexity.

I agree with Reviewer #2 that “the part of this work related to Wdr47 is complete and solid”, but I also agree that the portion of the manuscript related to the role of CAMSAPs and the role of Katanin presents some weaknesses. These are not fully addressed in the rebuttal precluding the manuscript from publication in Nature Communications in its current form.

I believe that the manuscript might be interesting enough for publication in Nature Communication in a heavily revised form, but the authors should modify the manuscript substantially both in the results and discussion sections to clarify the potentially different mechanistic roles of Camsap1 and 3 in CP formation and their role in cooperating with Katanin and keep the focus of their paper on Wdr47 data. I am sorry that my review cannot be more positive, as it is clear that the work required a major effort from the authors and already went through a round of review.

This reviewer is not convinced by the data suggesting that all Camsaps have a direct role in CP formation, they are CP proteins and co-localize with Wdr47 and found that area of the manuscript confusing.

Response:

We appreciate the comments and sincerely thank the reviewer for spending his/her precious time assessing the revised manuscript. As detailed below, we have responded to the comments to clarify the concerns of our reviewer. We have included new data and also modified the manuscript to further increase the strength and clarity of presentation and avoid overstatement.

- Camsaps show very different localization patterns in Fig.4c, some not convincingly consistent with a role in central pair formation. For example, Camsap3 is localized at the base of motile cilia as shown in Fig. 4b, away from Hydin a known central pair protein. How does Camsap3 play a role in CP formation when its subcellular localization is away from the CP region? Results presented previously in Robinson et al. PNAS June 2020 show that Camsap3 has a dual localization at basal bodies and

at the base of cilia (possibly the TZ), which is consistent with a role in CP formation. The localization data presented here seem different and not consistent with a role in CP formation.

In addition, Camsap1 also only shows minimal localization to the base of the CP in short cilia and is largely concentrated at the tip of growing motile cilia. How does this subpopulation of Camsap1 contribute to CP formation if at all? Assuming that only the small portion associated at the base of cilia of (some) of the Camsaps plays a role in CP formation, what kinds of role do the different Camsaps play in CP formation?

Response:

We are sorry for causing the confusion of our reviewer. Hydin is a component of the 2b projection along the C2 MT of CP (Fig. 2a) (Teves et al., 2016). CP still persists in Hydin-deficient mice (Lechtreck et al., 2008). Therefore, although Hydin is frequently used as a CP marker in the field, it is not a *bona fide* marker of central MTs. We have previously shown that the bottom area of CP contains a region (which we term as “CP-foot”) that is positive for Cep131 and Centrin but devoid of Hydin (Zhao et al., 2021), indicating that Hydin is excluded from the minus end region of the C2 MT. Furthermore, the length of CP-foot increases following the growth of cilia in mEPCs and can reach several hundred nanometers in length (Zhao et al., 2021). This explains why the Hydin-decorated CP region appears to be complementary to the Camsaps-decorated CP base (Fig. 4b). Concerning the localization of Camsaps relative to TZ, we have provided results to confirm that the Camsaps-decorated CP base region is above the plane of TZ, by using Cep290 and Cep162 as TZ markers (Fig. 4c, revised manuscript). These data further confirm that Camsaps are concentrated at the minus end region of CP, consistent with numerous EM studies showing that CP usually does not protrude into the TZ region. We have also included an illustration to aid understanding (Fig. 4c, revised manuscript). Therefore, our CP base localization of Camsap3 is consistent with and more precise than its ciliary base localization reported by Robinson and colleagues (Robinson et al., 2020). In the revised manuscript, we have modified the text accordingly and also emphasized that the Wdr47-decorated CP base region is also above TZ (Line 103) to increase the clarity of presentation.

Concerning their roles in CP formation, our results suggest that, although the ciliary localizations of Camsap1-3 are not identical, they share two common features. Firstly, in short cilia negative or weak for Hydin staining (thus still lacking CP or just initiating the CP assembly), Camsap1-3 are all highly concentrated at the cilia tip with Wdr47 (Fig. 1d,e; Fig. 4b-d, revised manuscript). Katanin-p60 is also similarly enriched at the tip of such short cilia (Fig. 7a,b). Together with other results (Fig. 5a-g, 6e-h, and 7d-g) and documented roles of Camsaps and Katanin, such as (Atherton et al., 2017; Dymek et al., 2004; Dymek and Smith, 2012; Hendershott and Vale, 2014; Jiang et al., 2018; Jiang et al., 2014; Sharma et al., 2007), we propose that the tip enrichment of these proteins facilitates their teamwork, i.e., generation of MT seeds by Katanin, stabilization of the seeds by Camsaps, and their appropriate targeting to the ciliary central lumen through Wdr47 (Fig. 7h). This model also echoes early studies in *C. reinhardtii* showing that central MTs initially emerge at the tip of short flagella and from a subdistal region of long flagella (Lechtreck et al., 2013).

Secondly, in long cilia, Camsap1-3 display prominent localizations at the bottom region of CP MTs (Fig. 4b-d). Wdr47 is also localized at the CP base and essential for both CP formation and maintenance (Fig. 1, 2). Therefore, based on the documented minus end-stabilizing properties of Camsaps (Akhmanova and Hoogenraad, 2015; Atherton et al., 2017; Hendershott and Vale, 2014; Jiang et al., 2014) and other results in the manuscript (Fig. 5,6), we propose that their CP-base enrichment stabilizes the minus-ends of CP MTs during CP assembly and maintenance (Fig. 7h).

Although our results suggest the involvement of all the Camsaps in the CP formation of mEPCs, the actual contribution of each Camsap appears to be affected by both its expression level and efficacy relative to the others (Fig. 5,6). Furthermore, emerging lines of evidence also support their importance in the CP formation of other tissue cells. For instance, Robinson and colleagues (Robinson et al., 2020) demonstrate that *Camsap3* deficiency only results in the CP loss in 77.4% of ciliary cross-sections of mouse airway epithelial cells. The animals display mild or no hydrocephalus phenotypes, implying a weaker effect of *Camsap3* deficiency on the CP formation of mEPCs. Two recent papers also report that, in *Camsap3*-deficient mice, ependymal multicilia show normal planar beat (Kimura et al., 2021) and oviduct multicilia contain normal CP (Usami et al., 2021). Future systematic analyses using single, double, and triple *Camsaps* gene knockout mice will be required to determine detailed contributions of individual Camsaps and their collective effects on the CP formation of various tissues.

- Data presented in Fig. 4c show Camsaps have different localization but somehow they always colocalize with Wdr47. The data presented in Figs. 4c and 7c are not convincing, the authors should swap the secondary antibodies colors of Ac-Tub (in blue) and the red channel (WDR47 for 4c and Camsap1 for 7c) to make sure there is no bleed through from GFP.

Response:

We appreciate these comments. When re-examining Figure 4c (now Fig. 4d in the revised manuscript), we realized that the two greyscale insets in the GFP-Camsap3 panel were both from the Wdr47 (red) channel. We have corrected this mistake in the revised manuscript and feel very sorry for this.

We have previously shown that all three Camsaps can associate with Wdr47. Furthermore, GFP-Camsap3 can recruit Wdr47 to MT minus ends in cells (Chen et al., 2020). The respective co-localization of Wdr47 with each of the GFP-tagged Camsaps in Figure 4C indicates that each of the GFP-Camsaps has the ability to recruit endogenous Wdr47. The results therefore further confirm the interplay between Wdr47 and Camsaps.

As to the bleed problem, we also hold the same concerns as our reviewer and thus have been careful on this in our research. In the revised manuscript, we have included Centrin1-GFP as negative control (Fig. 4d). The image contains two multiciliated cells differing markedly in expression levels of Centrin1-GFP. The fluorescent signals of Wdr47 and Centrin1-GFP, however, are independent of each other (Fig. 4d, revised manuscript), indicating no bleed between the two channels. In both cells, Wdr47 displays enriched

distributions at the tip of short cilia (Fig. 4d), consistent with our other results (e.g., Fig. 1d).

As the expression levels of GFP-Camsap1 were the highest among the three GFP-tagged Camsaps (Fig. 5a), we further take the image of GFP-Camsap1 in Figure 4c (Fig. 4d, revised manuscript) as another example. Reviewer Figure 1A shows a larger field of the original image from which the panel image in Figure 4c (Fig. 4d in the revised manuscript) was cropped (Cell-a). We can see that the neighboring cell (Cell-b) is positive for Wdr47 (Cy3-labeled) but negative for GFP-Camsap1. Both Cell-a and Cell-b, however, contain bright foci of Wdr47 at the tip of short cilia (arrows pointing to representative foci). The Ac-tubulin (Pacific blue-labeled) channel also shows no signs of bleed into other channels.

The fluorescent results in Figure 7b (our reviewer appeared to wrongly refer to Fig. 7c in the comment) were immunostained with the same set of secondary antibodies. As cells neighboring the one in Figure 7b are all positive for GFP-p60 in the original image, we herein use an image in different field as an example. Reviewer Figure 1B shows two multiciliated cells (Cell-c and Cell-d) with markedly different expression levels of GFP-p60 and another cell (Cell-e) negative for GFP-p60. As we can see, Camsap1 shows similar bright immunofluorescent signals at the tip of short cilia in all three cells. In addition, the images in Figure 5i provide additional examples to indicate that the fluorescent signals of Camsap1 are independent of those of GFP.

To further convince our reviewer, we also performed the suggested antibody-swap experiments. We expressed GFP-tagged Centrin1, Camsaps, and p60, respectively, and visualized endogenous Wdr47 or Camsap1 using an Alexa Fluor 647-conjugated secondary antibody. As shown in Reviewer Figure 1C-E, all the proteins still show similar localization patterns.

In addition, other multiplex fluorescent micrographs in the manuscript, such as those in Fig. 2e, Fig. 5d, Fig. 6f, and Fig. 7d, can also be used to exclude bleed problem between GFP channel and other channels.

- Overexpression of GFP-Camsap3 rescues only ~20% of the motile cilia for recruitment of Hydin in the WDR47 $-/-$ background, but rescue 75% of CP defects by TEM. Can the author look at other component of the CP for rescue in the GFP-Camsap3 overexpressing cells to check if this difference in the results is due to using different assays or due to a specific effect on HYDIN?

Response:

We realized that we used wrong colors when preparing the pie chart of the GFP-Camsap3 rescue results during the first round of revision. The rescue results are correctly described in the Result section (Line 254: “27% of axonemal cross-sections from the Camsap3 sample contained either one (1/173) or two (46/173) central MTs (Fig. 5g)”). We have corrected the colors of the pie chart in Figure 5g in the revised manuscript. Therefore, the actual percentage of cells with two CP MTs is 26.6%, which is consistent with the Hydin results. We are deeply sorry for the mistake.

- The authors show that Camsap1 and 3 overexpression convincingly rescue the phenotype on CP formation of Wdr47 $-/-$ cells. As pointed out by reviewer 2, GFP-Camsap2 rescue very mildly Wdr47 $-/-$ CP formation phenotype, so might have just an indirect role in CP formation driven by Wdr47.

As the authors suggest, there might be a certain degree of redundancy in the function of Camsaps such that overexpression might rescue some of the phenotype of Wdr47 caused by one of the Camsaps that is involved in CP formation. If this is correct, this work does not yet clarify which Camsap is relevant in CP formation. Alternatively, they might all play different roles in CP formation, but these different roles are not addressed.

Response:

Although the full-length Camsap2 showed a mild rescue effect, the LC and CKK constructs of Camsap2 display considerable rescue effects (Fig. 6e-g). As all the Camsap2-related constructs (GFP-Camsap2, GFP-Ca2LC, and GFP-Ca2CKK) showed low expression levels compared with the corresponding constructs of Camsap1 and Camsap3 (Fig. 5a; Fig. 6d), the rescue effects may at least partly be affected by the low expression levels. Furthermore, like Camsap1 and Camsap3, Camsap2 also localizes to the CP base (Fig. 4) and its MT minus end-binding CKK domain is also able to rescue the CP formation (Fig. 6), it unlikely plays an indirect role while the other two have direct roles. Camsap2 is also more homologous to Camsap3 in sequences than Camsap1 is (Akhmanova and Hoogenraad, 2015; Hendershott and Vale, 2014). Nevertheless, as discussed in our response to the first set of comments of our reviewer, lots of future investigations are still required to determine the detailed contributions of individual Camsaps and their collective effects on the CP formation.

As detailed in our responses thus far, current data suggest that all Camsaps are involved

in the CP formation because (1) they all localize to the tip of short cilia and the bottom region of CP (Fig. 4); (2) they display certain rescue effects when individually overexpressed in *Wdr47*-deficient mEPCs to increase the total levels of Camsaps (Fig. 5, 6); and (3) *Camsap3*-deficiency only abolishes CP in 77.4% of ciliary cross-sections of mouse airway epithelial cells and has mild effects on ependymal cilia and little effects on oviduct cilia (Kimura et al., 2021; Robinson et al., 2020; Usami et al., 2021). We agree with our reviewer that the detailed roles and contributions of individual Camsaps may not be identical. As this is expected to require systematic analyses using single, double, and triple *Camsap* gene knockout mice, we hope that our reviewer would agree that these detailed studies can be left to the future.

Our results suggest that *Wdr47* regionally enriches Camsaps of low physiological concentrations to ensure efficient and precise CP formation (Fig. 4, 5i-k, and 7h). *Wdr47* deficiency completely abolishes the CP formation (Fig. 1i; Supplementary Fig. 2; Fig. 7h). Increasing the total levels of Camsaps by overexpression partly compensates for the loss of *Wdr47*, rescuing CP formation with the expenses of low efficiency and abnormal CP number and position (Fig. 5, 6, 7h).

- The data on Katanin are interesting, but they don't justify the claims from the authors of an interplay between *Wdr47*, Camsaps and Katanin in CP formation. As discussed by Reviewer 2 "more experiments need to be conducted to explicitly state that Katanin cooperates with *Wdr47* and Camsaps for CP formation, as those stated in the title, the abstract, and the discussion..." The interplay statement should be toned down substantially in the manuscript and the discussion should clearly be focused on Camsap1 as the authors show only data for this protein. In the discussion the authors need to better elaborate on what is known about Camsap3 and Katanin interaction from published work on *Tetrahymena*. Ideally, they should compare in their system the roles of Camsap 1 and 3 in the interplay with Katanin function, even if the experiment would have some limitation as discussed in the rebuttal.

Response:

We thank our reviewer for appreciating our efforts.

Katanin has long been shown to be essential for the CP formation in protozoa (Dymek et al., 2004; Dymek and Smith, 2012; Sharma et al., 2007). Sharma and colleagues report that, in *Tetrahymena*, overexpressed katanin p60 selectively co-localizes with and severs outer but not central MTs in vivo. Furthermore, they propose a model that katanin severs outer doublet MTs to produce tubulin precursors for the CP (Sharma et al., 2007).

We find that increasing total Camsaps levels by overexpressing full-length Camsap1, Camsap3, or an LC or CKK construct of Camsaps in *Wdr47*-deficient mEPCs result in extra numbers of central MTs (Fig. 5f,g; Fig. 6g,h). Furthermore, the effects on central MT numbers show a positive correlation with the expression levels of the exogenous proteins (therefore the total levels of Camsaps) when similar constructs are compared (Fig. 5a; Fig.

6d). In a genetic background that completely lacks CP (Fig. 1i), these results are very striking, reflected not only in our presented results (Fig. 5f,g; Fig. 6g,h) but in original EM images as well (please refer to Reviewer Fig. 2, in which we have colored central MTs to facilitate examination). As central MTs are non-centrosomal MTs and Camsaps are known to function through stabilizing MT minus ends but not nucleating MTs (Akhmanova and Hoogenraad, 2015; Atherton et al., 2017; Hendershott and Vale, 2014; Jiang et al., 2014), these exciting observations strongly suggest that there are pre-existing MT seeds in the cilia and their stabilizations depend on Camsaps.

Reviewer Figure 2

Our data in Figure 7 demonstrated that (1) Katanin localizes in ependymal cilia and also shows enrichment at the tip of short cilia (Fig. 7a,b); (2) its MT-severing activity is important for the CP formation (Fig. 7c-f); and (3) its MT-severing activity is critical for GFP-Camsap1 overexpression-induced restoration of ciliary beat pattern in *Wdr47*-deficient mEPCs (Fig. 7g), suggesting that the central MTs rescued by elevated levels of total Camsaps also depend on Katanin. As tubulin dimers can be transported into cilia through intraflagellar transport (IFT) (Bhogaraju et al., 2013) and overexpression of GFP-p60 induced CP loss in a portion of ependymal axonemes (Fig. 7e,f), it is reasonable for us to modify the model by Sharma and colleagues (Sharma et al., 2007) and propose that Katanin provides Camsaps with central MT seeds by severing the periphery MT doublets (Fig. 7h).

As we stated in the previous rebuttal letter, we performed the experiments in Fig. 7g with GFP-Camsap1 because it is the best choice [“Firstly, in our hand GFP-Camsap1 displayed the highest expression levels and best rescue effects in *Wdr47*^{-/-} mEPCs (Fig. 5a-g). Therefore, it is obviously the best choice for further assessment of Katanin function. Secondly, because Camsap1 does not interact with Katanin, it allowed us to clearly attribute the effects of p60/p60^{K257A} to Katanin. The use of Camsap3 would complicate the situation. For instance, we would need to confirm first that p60 and p60^{K257A} would not display different binding affinity to Camsap3 and differentially affect its MT-binding and stabilizing activities.”]. As the reviewer still expresses concerns on the interplay between Katanin and Camsaps, we

performed additional experiments using GFP-Ca3LC. This Camsap3 deletion mutant is fairly expressed (Fig. 6d), displays prominent CP rescue efficiency (Fig. 6e-h), and lacks the katanin-binding (KB) domain (Fig. 6c) defined by Jiang and colleagues (Jiang et al., 2018). We co-expressed GFP-Ca3LC with RFP-p60^{K257A} or Centrin1-RFP (negative control) in *Wdr47*^{-/-} mEPCs and analyzed ciliary beat patterns. As shown in Figure 7g in the revised manuscript, p60^{K257A} also markedly repressed the rescue effects of Ca3LC. These results again suggest that the central MT seeds stabilized by elevated levels of total Camsaps derive primarily from Katanin-mediated MT severing.

Therefore, our results collectively suggest that *Wdr47*, Camsaps, and Katanin function as a team in which Katanin generates MT seeds, Camsaps stabilize the seeds, and *Wdr47* concentrates Camsaps to appropriate ciliary locations for efficient, proper CP formation. We agree with our reviewer that more experiments are required to reveal detailed interplay among Katanin, Camsaps, and *Wdr47*. Such studies, however, will require rescue experiments performed in multiciliated cells lacking all three Camsaps plus *Wdr47*. We thus hope that our reviewer would agree that these issues can await future investigations. During the revision, we have modified the manuscript to increase the clarity of presentation and avoid overstatement. We are sorry that we cannot find the publication reporting the Camsap3-Katanin interaction in *Tetrahymena* to fulfill the request of our reviewer. To the best of our knowledge, protozoan Camsap has not been identified. Nonetheless, we have included more discussions on previous works on protozoan Katanin. We wish our reviewer recognize our tremendous efforts in the studies trying to achieve a relatively comprehensive understanding of mechanisms underlying mammalian CP formation.

References

- Akhmanova, A., and Hoogenraad, C.C. (2015). Microtubule minus-end-targeting proteins. *Curr Biol* *25*, R162-171.
- Atherton, J., Jiang, K., Stangier, M.M., Luo, Y., Hua, S., Houben, K., van Hooff, J.J.E., Joseph, A.P., Scarabelli, G., Grant, B.J., *et al.* (2017). A structural model for microtubule minus-end recognition and protection by CAMSAP proteins. *Nat Struct Mol Biol* *24*, 931-943.
- Bhogaraju, S., Cajanek, L., Fort, C., Blisnick, T., Weber, K., Taschner, M., Mizuno, N., Lamla, S., Bastin, P., Nigg, E.A., *et al.* (2013). Molecular basis of tubulin transport within the cilium by IFT74 and IFT81. *Science* *341*, 1009-1012.
- Chen, Y., Zheng, J., Li, X., Zhu, L., Shao, Z., Yan, X., and Zhu, X. (2020). *Wdr47* Controls Neuronal Polarization through the Camsap Family Microtubule Minus-End-Binding Proteins. *Cell Rep* *31*, 107526.
- Dymek, E.E., Lefebvre, P.A., and Smith, E.F. (2004). PF15p is the chlamydomonas homologue of the Katanin p80 subunit and is required for assembly of flagellar central microtubules. *Eukaryot Cell* *3*, 870-879.
- Dymek, E.E., and Smith, E.F. (2012). PF19 encodes the p60 catalytic subunit of katanin and is required for assembly of the flagellar central apparatus in *Chlamydomonas*. *J Cell Sci* *125*, 3357-

3366.

- Hendershott, M.C., and Vale, R.D. (2014). Regulation of microtubule minus-end dynamics by CAMSAPs and Patronin. *Proc Natl Acad Sci U S A* *111*, 5860-5865.
- Jiang, K., Faltova, L., Hua, S., Capitani, G., Prota, A.E., Landgraf, C., Volkmer, R., Kammerer, R.A., Steinmetz, M.O., and Akhmanova, A. (2018). Structural Basis of Formation of the Microtubule Minus-End-Regulating CAMSAP-Katanin Complex. *Structure* *26*, 375-382 e374.
- Jiang, K., Hua, S., Mohan, R., Grigoriev, I., Yau, K.W., Liu, Q., Katrukha, E.A., Altelaar, A.F., Heck, A.J., Hoogenraad, C.C., *et al.* (2014). Microtubule minus-end stabilization by polymerization-driven CAMSAP deposition. *Dev Cell* *28*, 295-309.
- Kimura, T., Saito, H., Kawasaki, M., and Takeichi, M. (2021). CAMSAP3 is required for mTORC1-dependent ependymal cell growth and lateral ventricle shaping in mouse brains. *Development* *148*.
- Lechtreck, K.F., Delmotte, P., Robinson, M.L., Sanderson, M.J., and Witman, G.B. (2008). Mutations in Hydin impair ciliary motility in mice. *J Cell Biol* *180*, 633-643.
- Lechtreck, K.F., Gould, T.J., and Witman, G.B. (2013). Flagellar central pair assembly in *Chlamydomonas reinhardtii*. *Cilia* *2*, 15.
- Robinson, A.M., Takahashi, S., Brotslaw, E.J., Ahmad, A., Ferrer, E., Procissi, D., Richter, C.P., Cheatham, M.A., Mitchell, B.J., and Zheng, J. (2020). CAMSAP3 facilitates basal body polarity and the formation of the central pair of microtubules in motile cilia. *Proc Natl Acad Sci U S A*.
- Sharma, N., Bryant, J., Wloga, D., Donaldson, R., Davis, R.C., Jerka-Dziadosz, M., and Gaertig, J. (2007). Katanin regulates dynamics of microtubules and biogenesis of motile cilia. *J Cell Biol* *178*, 1065-1079.
- Teves, M.E., Nagarkatti-Gude, D.R., Zhang, Z., and Strauss, J.F., 3rd (2016). Mammalian axoneme central pair complex proteins: Broader roles revealed by gene knockout phenotypes. *Cytoskeleton (Hoboken)* *73*, 3-22.
- Usami, F.M., Arata, M., Shi, D.B., Oka, S., Higuchi, Y., Tissir, F., Takeichi, M., and Fujimori, T. (2021). Intercellular and intracellular cilia orientation is coordinated by CELSR1 and CAMSAP3 in oviduct multi-ciliated cells. *J Cell Sci* *134*.
- Zhao, H., Chen, Q., Li, F., Cui, L., Xie, L., Huang, Q., Liang, X., Zhou, J., Yan, X., and Zhu, X. (2021). Fibrogranular materials function as organizers to ensure the fidelity of multiciliary assembly. *Nat Commun* *12*, 1273.

Reviewers' Comments:

Reviewer #3:

Remarks to the Author:

The authors addressed requests of this reviewer appropriately. I recommend the manuscript for publication in the current form.